# Repression of developmental transcription factor networks triggers aging-associated gene expression in human glial progenitor cells

John N. Mariani[1] ✉, Benjamin Mansky[1], Pernille M. Madsen[1,2], Dennis Salinas[1], Deniz Kesmen[1], Nguyen P. T. Huynh[2], Nicholas J. Kuypers[1], Erin R. Kesel[1], Janna Bates[1], Casey Payne[1], Devin Chandler-Militello[1], Abdellatif Benraiss[1] & Steven A. Goldman ![ORCID][1,2] ✉

Human glial progenitor cells (hGPCs) exhibit diminished expansion competence with age, as well as after recurrent demyelination. Using RNA-sequencing to compare the gene expression of fetal and adult hGPCs, we identify age-related changes in transcription consistent with the repression of genes enabling mitotic expansion, concurrent with the onset of aging-associated transcriptional programs. Adult hGPCs develop a repressive transcription factor network centered on MYC, and regulated by ZNF274, MAX, IKZF3, and E2F6. Individual over-expression of these factors in iPSC-derived hGPCs lead to a loss of proliferative gene expression and an induction of mitotic senescence, replicating the transcriptional changes incurred during glial aging. miRNA profiling identifies the appearance of an adult-selective miRNA signature, imposing further constraints on the expansion competence of aged GPCs. hGPC aging is thus associated with acquisition of a MYC-repressive environment, suggesting that suppression of these repressors of glial expansion may permit the rejuvenation of aged hGPCs.

Glial progenitor cells (GPCs, also referred to as oligodendrocyte progenitor cells or NG2 cells) emerge during the 2nd trimester to colonize the human brain, and persist in abundance throughout adulthood. During development, human GPCs (hGPCs) are highly proliferative bipotential cells, producing new oligodendrocytes and astrocytes[1,2]. In rodents, this capacity wanes during normal aging, with proliferation, migration, and differentiation competence all diminishing in aged GPCs[3–11]. Similarly, adult human GPCs are less proliferative, less migratory, and more readily differentiated than their fetal counterparts when transplanted into neonatal murine hosts[12]. Yet despite the manifestly different competencies of fetal and adult hGPCs, and the

abundant data on GPC transcription in rodent models of aging[13–17], little data are available that address changes in GPC gene expression during normal human aging[18–21]. We therefore sought to compare the transcriptional patterns of fetal and adult hGPCs, and to use that data to identify those regulatory pathways causally linked to the maturation and aging of these cells.

To this end, we utilized bulk and single cell RNA-Sequencing (scRNA-Seq) of hGPCs isolated from human fetal forebrain, so as to define their transcriptional signatures and heterogeneity. We then compared these data to the gene expression of isolated adult hGPCs, and found that the latter exhibited transcriptional patterns suggesting

[1]Center for Translational Neuromedicine, University of Rochester Medical Center, Rochester, NY 14642, USA. [2]Center for Translational Neuromedicine, University of Copenhagen Faculty of Health, Copenhagen 2200, Denmark. ✉e-mail: John_mariani@urmc.rochester.edu; Steven_goldman@urmc.rochester.edu

a loss of proliferative capacity, the onset of an early phenotypically-differentiated profile, and the induction of senescence. Transcription factor motif enrichment analysis of the promoters of differentially expressed genes then implicated the adult-induced transcriptional repressors E2F6, ZNF274, MAX, and IKZF3 as principal drivers of the

human glial aging program. Network analysis strongly suggested that as a group, these genes worked though the inhibition of MYC and its proximal targets, which were relatively over-expressed in fetal hGPCs. We found that over-expression of these adult repressors in newly generated human iPSC-derived GPCs, which are analogous to fetal

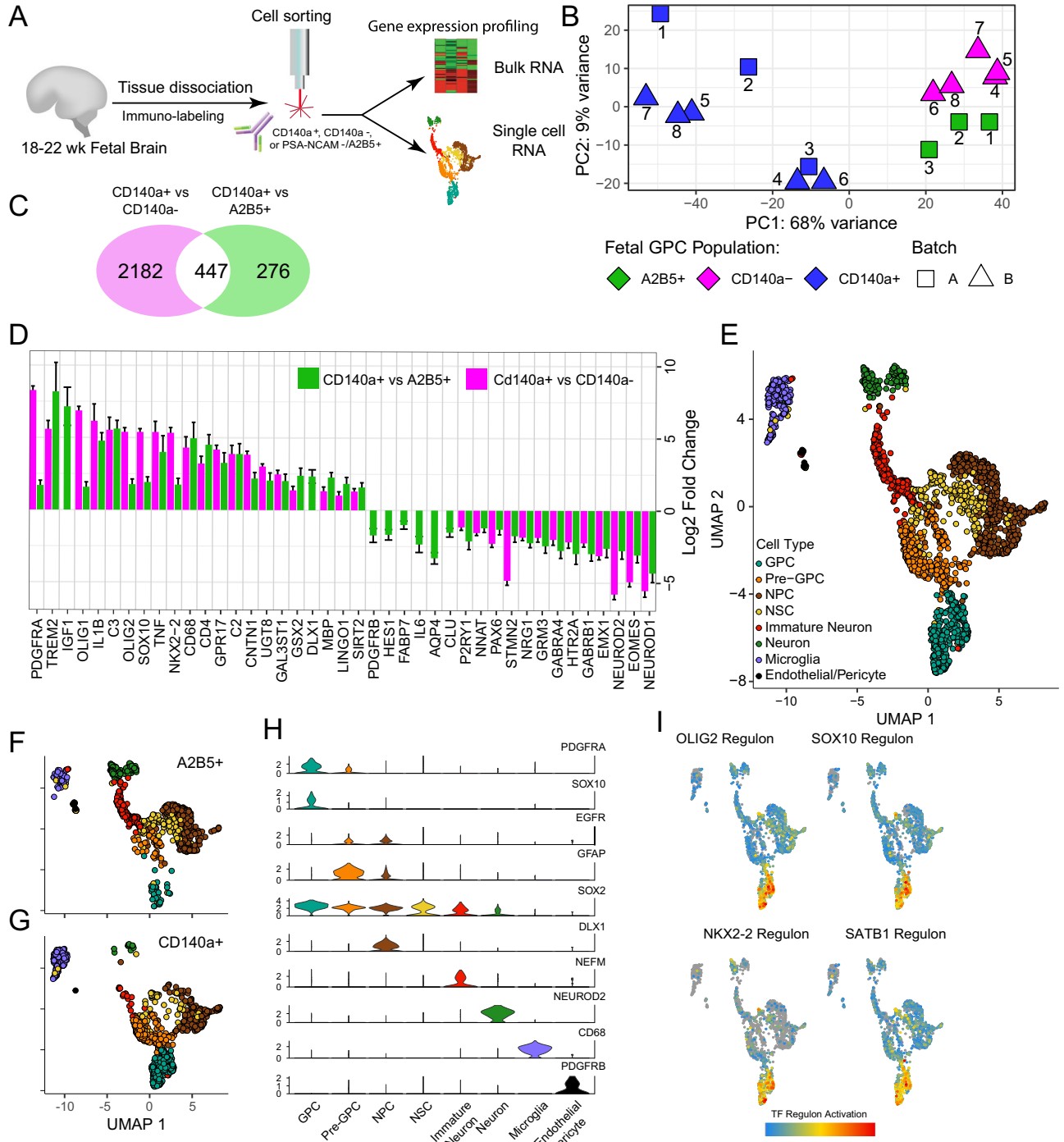

**Fig. 1 | Transcriptomic characterization of FACS-isolated human fetal GPCs.**
**A** Workflow of bulk and scRNA-Sequencing of CD140a⁺, CD140a⁻, and A2B5⁺/PSA-NCAM⁻-selected 2nd trimester human fetal brain isolates. **B** Principal component analysis of all samples across two batches. Numbers indicate pairing of samples (fetal CD140⁺ $n$ = 8; CD140a⁻, $n$ = 5; A2B5⁺/PSA-NCAM⁻, $n$ = 3; biologically independent samples). **C** Venn diagram of CD140a⁺ vs CD140a⁻ and CD140⁺ vs A2B5⁺/PSA-NCAM⁻ differentially-expressed gene sets (FDR < 0.01 and absolute log₂-fold change > 1, calculated with DESeq2). **D** Log₂-fold changes of significant curated

genes for each geneset. Missing bars were not significant. **E** UMAP plot of the primary cell types identified during scRNA-Seq analysis of all FACS-isolated hGPCs derived from 20 week gestational age human fetal VZ/SVZ. **F-G** UMAPs of PSA-NCAM⁻/A2B5⁺ (**F**) vs. CD140a⁺ (**G**) human fetal cells. **H** Violin plots of cell type-selective marker genes. **I** Select feature plots of transcription factor regulons (calculated in SCENIC), predicted to be significantly activated in fetal hGPCs (FDR < 0.01, Wilcoxon rank-sum test). Source data are provided as a Source Data file. Error bars ± SEM.

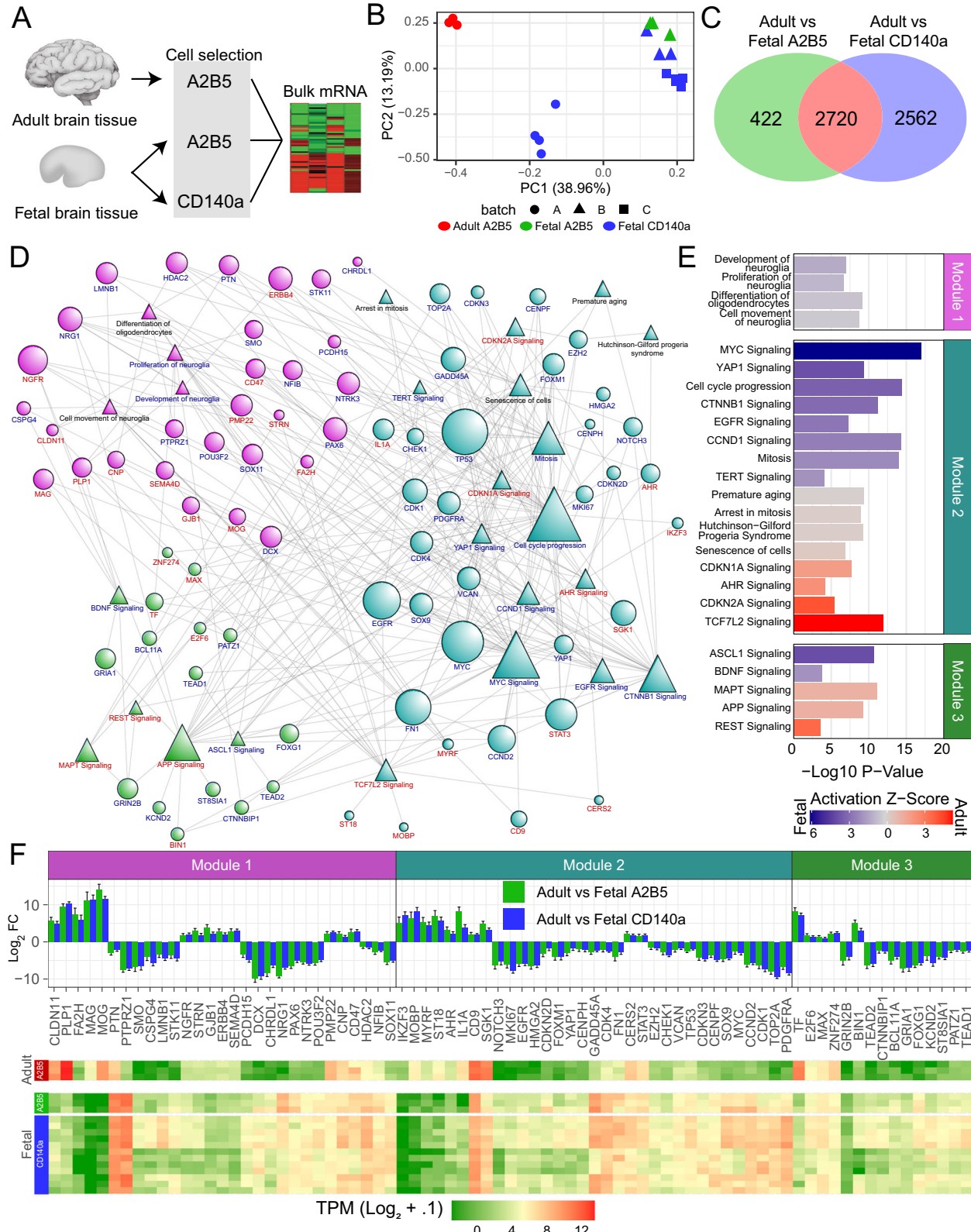

**Fig. 2 | Adult human GPCs are transcriptionally and functionally distinct from fetal hGPCs. A** Workflow of bulk RNA-Seq analysis of human adult and fetal GPCs. **B** Principal component analysis of all samples across three batches (fetal CD140a+, $n = 12$; fetal A2B5+/PSA-NCAM-, $n = 3$; adult A2B5+, $n = 3$; biologically independent samples). **C** Venn Diagram of Adult vs Fetal differentially expressed gene sets. (FDR < 0.01, Log2FC > 1, calculated with DESeq2). **D** IPA network of curated significant terms and genes (FDR < 0.001). Node size is proportionate to node degree. Color corresponds to enrichment in either adult (*red*) or fetal (*blue*) populations. **E** Bar plots of significant IPA terms by module. Z-Scores indicate predicted activation in fetal (*blue*) or adult (*red*) hGPCs. **F** Log2-fold changes and heatmap of network gene TPMs. Source data are provided as a Source Data file. Error bars ± SEM.

hGPCs in their expression signatures, induced transcriptional signatures that substantially recapitulated those of adult hGPCs. We then identified a cohort of miRNAs selectively-expressed by adult hGPCs, whose targets predicted the post-transcriptional inhibition of fetal hGPC gene expression, especially so in concert with the adult-acquired repressor network. Together, these data suggest that a cohort of repressors appears during the aging of adult human GPCs, whose activity is centered on MYC and MYC-dependent transcription. As such, these repressors may comprise feasible therapeutic targets, whose modulation may restore salient features of the mitotic and differentiation competence of aged or otherwise mitotically-exhausted hGPCs.

## Results

### CD140a selection enriches most specifically for human fetal glial progenitors

To identify the transcriptional concomitants to hGPC aging, we first used bulk and single cell RNA-Seq to characterize hGPCs derived from second trimester fetal human tissue, whether isolated by targeting the CD140a epitope of PDGFRα[22], or the glial gangliosides recognized by monoclonal antibody A2B5[12,22,23]. Since A2B5 also recognizes young neurons in early development, we used two-color FACS to immuno-deplete fetal brain samples for PSA-NCAM at the time of A2B5 enrichment, so as to exclude contaminating neuroblasts[12]; hence the definition of this fetal hGPC pool as A2B5+/PSA-NCAM-. On that basis, sample-matched experiments were carried out whereby the ventricular/subventricular zones (VZ/SVZ) of 18–22 week gestational age (g.a.) fetal brains were dissociated and sorted via fluorescence activated cell sorting (FACS), for either CD140a+ or A2B5+/PSA-NCAM- hGPCs ($n = 3$, as matched isolates from the same fetal brains), or for CD140a+ hGPCs as well as the CD140a-depleted remainder ($n = 5$; Fig. 1A). Bulk RNA-Seq libraries were then generated and deeply sequenced for both experiments. Principal component analysis (PCA) showed segregation of the CD140a+ and A2B5+ cells, and further segregation of both from the CD140a-depleted samples (Fig. 1B). Differential expression in both paired cohorts ($p < 0.01$, absolute log$_2$ fold change >1) identified 723 genes as differentially-expressed between CD140a+ and A2B5+ hGPCs (435 in CD140a, 288 in A2B5, Supplementary Data 1). In contrast, 2,629 genes distinguished CD140a+ hGPCs from CD140a- (Fig. 1C; Supplementary Fig. 1A; Supplementary Data 1). Differential gene expression directionality was highly consistent when comparing CD140+ to either A2B5+ or CD140- cells, with all but 4 genes concordant (Supplementary Data 1).

Pathway enrichment analysis using Ingenuity Pathway Analysis (IPA) of both of these gene sets identified similar pathways as relatively active in hGPCs; these pathways included cell movement, oligodendroglial differentiation, lipid synthesis, and downstream PDGF, SOX10, and TCF7L2 signaling (Supplementary Fig. 1B, Supplementary Data 1). As expected, stronger activation Z-scores were typically observed when comparing CD140a+ hGPCs to CD140a- cells rather than to A2B5+ hGPCs. Among the genes differentially upregulated in CD140a+ isolates were PDGFRA itself, and a number of early oligodendroglial genes including OLIG1, OLIG2, NKX2-2, SOX10, and GPR17 (Fig. 1D). Furthermore, the CD140a+ fraction also exhibited increased expression of later myelinogenesis-associated genes, including MBP, GAL3ST1, and UGT8. Beyond enrichment of the oligodendroglial lineage, many genes typically associated with microglia and immune activation were also enriched in the CD140a isolates, including CD68, C2, C3, CD4, and TREM2; this was at least in part likely due to the inclusion of microglia following their re-expression of phagocytosed PDGFαR epitopes. In contrast, A2B5+ isolates exhibited enrichment of astrocytic (AQP4, CLU) and early neuronal (NEUROD1, NEUROD2, GABRG1, GABRA4, EOMES, HTR2A) genes, suggesting the expression of A2B5 by immature astrocytes and neurons as well as by hGPCs and oligodendroglial lineage cells. Overall then, oligodendroglial enrichment was greater in

CD140a+ hGPCs than A2B5-defined hGPCs, when each was compared to depleted fractions, suggesting the CD140a isolates as being the more enriched in hGPCs, and thus CD140a as the more appropriate phenotype for head-to-head comparison with adult hGPCs.

To further define the composition of fetal hGPC isolates at single cell resolution, we isolated both CD140a+ and A2B5+ hGPCs from 20 week g.a. fetal VZ/SVZ via FACS, and then assayed the transcriptomes of each by single cell RNA-Seq (Fig. 1A, 10X Genomics V2). We sought to capture ≥1000 cells of each; following filtration of low-quality cells (low-quality = unique genes < 500 or mitochondrial gene percentage > 15%), we were left with 1053 A2B5+/PSA-NCAM- and 957 CD140a+ high quality cells (median 6845 unique molecular identifiers and 2336 unique genes per cell). Dimensionality reduction via uniform manifold approximation and projection (UMAP), followed by shared nearest neighbor modularity-based clustering of all cells using Seurat[24], revealed 11 clusters with 8 primary cell types, as defined by their differential enrichment of marker genes (Supplementary Data 2). The primary cell types included: GPCs, pre-GPCs, neural stem and progenitor cells (NSCs and NPCs), immature neurons, neurons, microglia, and a cluster consisting of endothelial cells and pericytes (Fig. 1E). Consistent with our bulk RNA-Seq analysis, single cell RNA-Seq confirmed that the CD140a+ FACS isolates were more enriched for GPC and pre-GPC populations than were the fetal A2B5+/PSA-NCAM- cells (Fig. 1E–H, Supplementary Fig. 1C, D). Furthermore, whereas the CD140a-sorted cells were largely limited to GPCs and pre-GPCs, with only scattered microglial contamination, the A2B5+/PSA-NCAM- isolates also included astrocytes and neuronal lineage cells, the latter despite the upfront depletion of neuronal. These data supported the more selective and phenotypically-restricted nature of CD140a rather than A2B5-based hGPC isolation.

On that basis, we next explored the gene expression profiles of the predominant cell populations in the CD140a+ fetal isolates, GPCs and pre-GPCs[25]. Differential expression between these two pools yielded 269 (143 upregulated, 126 downregulated; $p < 0.01$, log2 fold change > 0.5; Supplementary Fig. 1E, Supplementary Data 2). During the pre-GPC to GPC transition, early oligodendroglial lineage genes were rapidly upregulated (OLIG2, SOX10, NKX2-2, PLLP, APOD), whereas those expressed in pre-GPCs effectively disappeared (VIM, HOPX, TAGLN2, TNC). Interestingly, genes involved in the human leukocyte antigen system, including HLA-A, HLA-B, HLA-C and B2M, were all *downregulated* as the cells transitioned to the GPC stage (Supplementary Fig. 1F, Supplementary Data 2). IPA analysis indicated that pre-GPCs were relatively enriched for terms related to migration, proliferation, and those presaging astrocytic identity (BMP4, AGT, and VEGF signaling), whereas GPCs were enriched for terms associated with acquisition of an oligodendroglial identity (PDGF-AA, FGFR2, CCND1), in addition to activation of the MYC and MYCN pathways (Supplementary Fig. 1G, Supplementary Data 2). Using single cell co-expression data together with promoter motif enrichment via SCENIC[26], we then identified 262 transcription factors as predicted to be relatively activated in GPCs vs pre-GPCs (Wilcoxon test, FDR < 0.01, Supplementary Data 2). These included SATB1, as well as the early GPC specification factors OLIG2, SOX10, and NKX2-2 (Fig. 1I).

### Human adult and fetal GPCs are transcriptionally distinct

We next asked how adult hGPCs might differ transcriptionally from fetal hGPCs. To this end, A2B5+ hGPCs were isolated from surgically-resected adult human temporal neocortex (18–27 years old, $n = 3$) and their bulk RNA sequenced together with 4 additional fetal CD140a+ samples, enabling regression of sequencing batch effects while simultaneously increasing power (Fig. 2A). We have previously noted that A2B5 selection is sufficient to isolate GPCs from adult human brain[18], and is more sensitive in adults than selection based on CD140a/PDGFαR, since PDGFRA expression falls with maturation and is typically lowly expressed by adult hGPCs[12,14,18,20,27]. Indeed, we found that

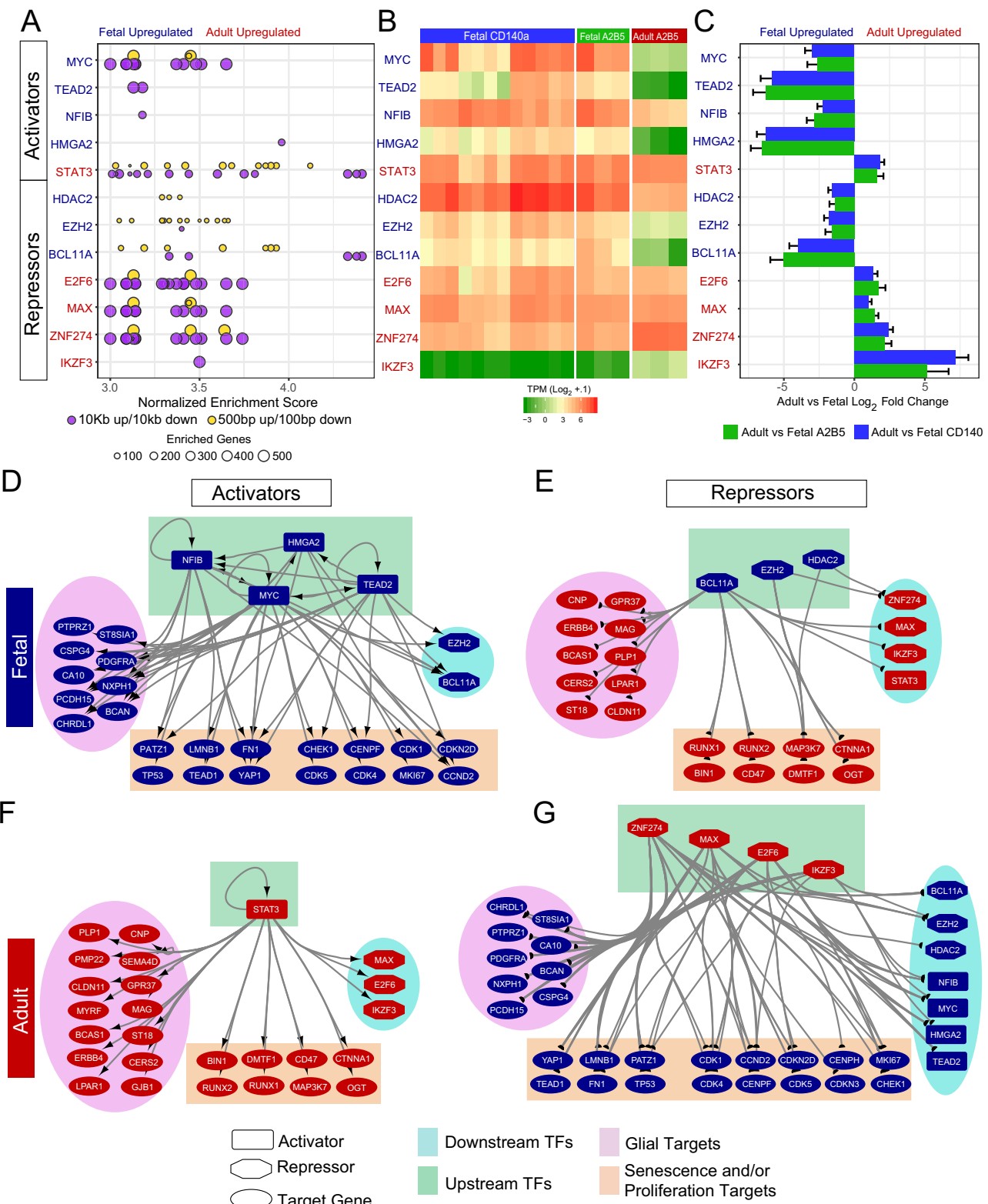

**Fig. 3 | Inference of transcription factor activity implicates a set of transcriptional repressors in the establishment of adult hGPC identity. A** Normalized enrichment score plots of significantly enriched transcription factors predicted to be active in fetal and adult GPCs. Each dot is a motif whose size indicates how many genes in which that motif is predicted to be active; color represents the window around the promoter at which that motif was enriched. **B** Heatmap of enriched TF TPMs, and (**C**) log-fold changes for both fetal hGPC isolates vs adult GPCs (fetal CD140a⁺, n = 12; fetal A2B5⁺/PSA-NCAM⁻, n = 3; adult A2B5⁺, n = 3; biologically

independent samples). **D–G** Predicted direct transcription factor activity of curated genes split into (**D**) fetal activators; (**E**) fetal repressors; (**F**) adult activators; and (**G**) adult repressors. Color indicates differential expression in adult (*red*) or fetal (*blue*) hGPCs; shape dictates type of node (*octagon*, repressor; *rectangle*, activator; *oval*, other target gene). Boxed and circled genes indicate functionally-related genes contributing to either glial progenitor/oligodendrocyte identity, senescence/proliferation targets, or upstream or downstream TFs that were also deemed activated. Source data are provided as a Source Data file. Error bars ± SEM.

PDGFRA in adult A2B5[+] hGPCs was expressed with a median TPM of only 0.55, compared to its median TPM of 47.56 in fetal A2B5[+] cells accordingly, few PDGFaR+ cells could be identified cytometrically or be isolated from adult brain tissue. Furthermore, depletion of PSA-NCAM[+] cells was not necessary for adult hGPC samples, as the expression of PSA-NCAM ceases in neurons of the adult cortex and white matter[28]. As a result, principal component analysis (PCA) showed tight clustering of adult hGPCs, that sharply segregated from both fetal hGPC pools (Fig. 2B). Differential expression of adult hGPCs compared to either CD140a[+] or A2B5[+] fetal hGPCs yielded 5282 and 3142 significant genes, respectively ($p < 0.01$; absolute log$_2$ fold-change > 1) (Fig. 2C, Supplementary Data 3). Downstream analyses were then carried out on the intersecting 2720 genes (1060 upregulated and 1660 downregulated in adult relative to fetal hGPCs). Remarkably, within these differentially-expressed gene sets, 100% of genes were directionally concordant.

To better understand the differences between adult and fetal hGPCs, we next constructed a network of non-redundant significant IPA terms and their contributing differentially-expressed genes (Fig. 2D, E). Spin glass community detection of this network[29] uncovered three modules (Modules M1-M3) of highly connected functional terms and genes (Fig. 2F, Supplementary Data 3). Module M1 included terms and genes linked to glial development, proliferation, and movement. Notably, a number of genes associated with GPC ontogeny were downregulated in adult hGPCs; these included CSPG4/NG2, PCDH15, CHRDL1, LMNB1, PTPRZ1, and ST8SIA1[25,30-33]. In contrast, a number of genes whose appearance precedes and continues through oligodendrocyte differentiation and myelination were upregulated in adult hGPCs, including MAG, MOG, MYRF, PLP1, CD9, CLDN11, CNP, ERBB4, GJB1, PMP22, and SEMA4D.

Module M2 harbored numerous terms associated with cellular aging and the modulation of proliferation and senescence. Cell cycle progression and mitosis were predicted to be activated in fetal hGPCs due to strong enrichment of proliferative factors including MKI67, TOP2A, CENPF, CENPH, CHEK1, EZH2 and numerous cyclins, including CDK1 and CDK4. Furthermore, proliferation-associated pathways were also inferred to be activated; these included MYC, CCND1, and YAP1 signaling, of which both YAP1 and MYC transcripts were similarly upregulated[34-36]. In that regard, transient overexpression of MYC in aged rodent GPCs has recently been shown to restore their capacity to both proliferate and differentiate[37]. In contrast, adult hGPCs exhibited an upregulation of senescence-associated transcripts, including E2F6, MAP3K7, DMTF1/DMP1, OGT, AHR, RUNX1, and RUNX2[38-42]. At the same time, adult hGPCs exhibited a downregulation of fetal transcripts that included LMNB1, PATZ1, BCL11A, HDAC2, FN1, EZH2, and both YAP1 and its cofactor TEAD1[36,43-48]. As a result, functional terms predicted to be active in adult hGPCs included senescence, the rapid onset of aging observed in Hutchinson-Gilford progeria, and cyclin-dependent kinase inhibitory pathways downstream of CDKN1A/p21 and CDKN2A/p16. Furthermore, AHR and its signaling pathway, which has been implicated in driving senescence via the inhibition of MYC[49], was similarly upregulated in adult GPCs.

Module M3 consisted primarily of developmental and disease linked signaling pathways that have also been associated with aging. These included the predicted activation of ASCL1 and BDNF signaling in fetal hGPCs, and MAPT/Tau, APP, and REST signaling in adult hGPCs[50-52]. Overall, the transcriptional and functional profiling of adult hGPCs revealed a reduction in transcripts associated with proliferative capacity, and a shift toward senescence and more mature phenotype.

## Inference of transcription factor activity implicates adult hGPC transcriptional repressors

Given the significant transcriptional disparity between adult and fetal GPCs, we next asked whether we could infer which transcription

factors (TFs) direct their identities. To accomplish this, we first scanned two promoter windows (500 bp up/100 bp down, 10 kb up/10 kb down) of adult or fetal enriched hGPC gene sets to infer significantly enriched TF motifs[26]. This identified 48 TFs that were also differentially-expressed in the scanned intersecting dataset (Supplementary Fig. 2, Supplementary Data 3). Among these, we focused on TFs whose primary means of DNA interaction were exclusively either repressive or stimulatory, while also considering the enrichment of their known cofactors. This analysis yielded 12 potential upstream regulators to explore (Fig. 3A-C): 4 adult repressors, E2F6, ZNF274, MAX, and IKZF3; 1 adult activator, STAT3; 3 fetal repressors, BCL11A HDAC2, and EZH2; and 4 fetal activators, MYC, HMGA2, NFIB, and TEAD2. Interestingly, of these predicted TFs, 3 groups shared a high concordance of motif similarity within their targeted promoters: (1) E2F6, ZNF274, MAX, and MYC; (2) STAT3 and BCL11A; and (3) EZH2 and HDAC2, suggesting that they may cooperate or compete for DNA binding at shared loci (Fig. 3A and Supplementary Fig. 2).

We next constructed four predicted signaling networks based on curated transcriptional interactions, to predict those genes targeted by our set of TFs (Fig. 3D-G). Among activators enriched in fetal GPCs (Fig. 3D), MYC, a proliferative factor[53], NFIB, a key determinant of gliogenesis[54], TEAD2, a YAP/TAZ effector, and HMGA2, a developmental regulator of chromatin architecture, were each predicted to activate cohorts of progenitor stage genes, including both mitogenesis-associated transcripts, and those demonstrated to inhibit the onset of senescence[53-56]. Direct positive cross-regulation was also predicted among these four fetal activators, with NFIB being driven by HMGA2 and TEAD2, MYC being driven by TEAD2 and NFIB, HMGA2 being driven by MYC and TEAD2, and TEAD2 being reciprocally driven by MYC (Fig. 3D). In contrast to these fetal activators, fetal stage repressors, including the C2H2 type zinc finger BCL11A, the polycomb repressive complex subunit EZH2, and histone deacetylase HDAC2, were each predicted to repress more mature oligodendrocytic gene expression at this stage (Fig. 3E)[57-59]. Furthermore, all three of these latter transcripts – BCL11A, EZH2 and HDAC2 - were predicted to inhibit targets implicated in senescence. As such, these factors appear to directly orchestrate downstream transcriptional events leading to maintenance of the cycling progenitor state.

We next sought to identify a mechanism responsible for these age-related changes in gene expression. STAT3 was predicted to shift GPC identity towards glial maturation via the upregulation of a large cohort of early differentiation- and myelination-associated oligodendrocytic genes (Fig. 3F). In addition, STAT3 was also inferred to activate a set of senescence-associated genes including BIN1, RUNX1, RUNX2, DMTF1, CD47, MAP3K7, CTNNA1, and OGT. At the same time, repression in adult hGPCs was predicted to be effected through the Ikaros family zinc finger IKZF3/Aiolos, the KRAB (kruppel associated box) zinc finger ZNF274, the MYC-associated factor MAX, and the cell cycle regulator E2F6 (Fig. 3G)[60-63]. Targeting by this set of transcription factors – IKZF3, ZNF274, MAX and E2F6 - predicted repression of those gene sets contributing to the fetal hGPC signature, and this was indeed observed in the downregulation of the early progenitor genes PDGFRA and CSPG4, as well as of the cell cyclicity genes CDK1, CDK4, and MKI67. Repression by these TFs of YAP1, LMNB1, and TEAD1 - whose expression is associated with developmental growth and population expansion - was also predicted. Interestingly, the expression of these four adult repressors predicted not only the downregulation of the fetal-enriched activators, NFIB, MYC, TEAD2 and HMGA2, but also that of their own negative regulators, the fetal repressors BCL11A, EZH2, and HDAC2 - thus potentiating maintenance of a senescent state once achieved.

## Expression of adult-enriched repressors induces age-associated changes in GPCs

On the basis of these findings, we asked if the adult repressors - E2F6, IKZF3, MAX and ZNF274 - were individually sufficient to induce aspects

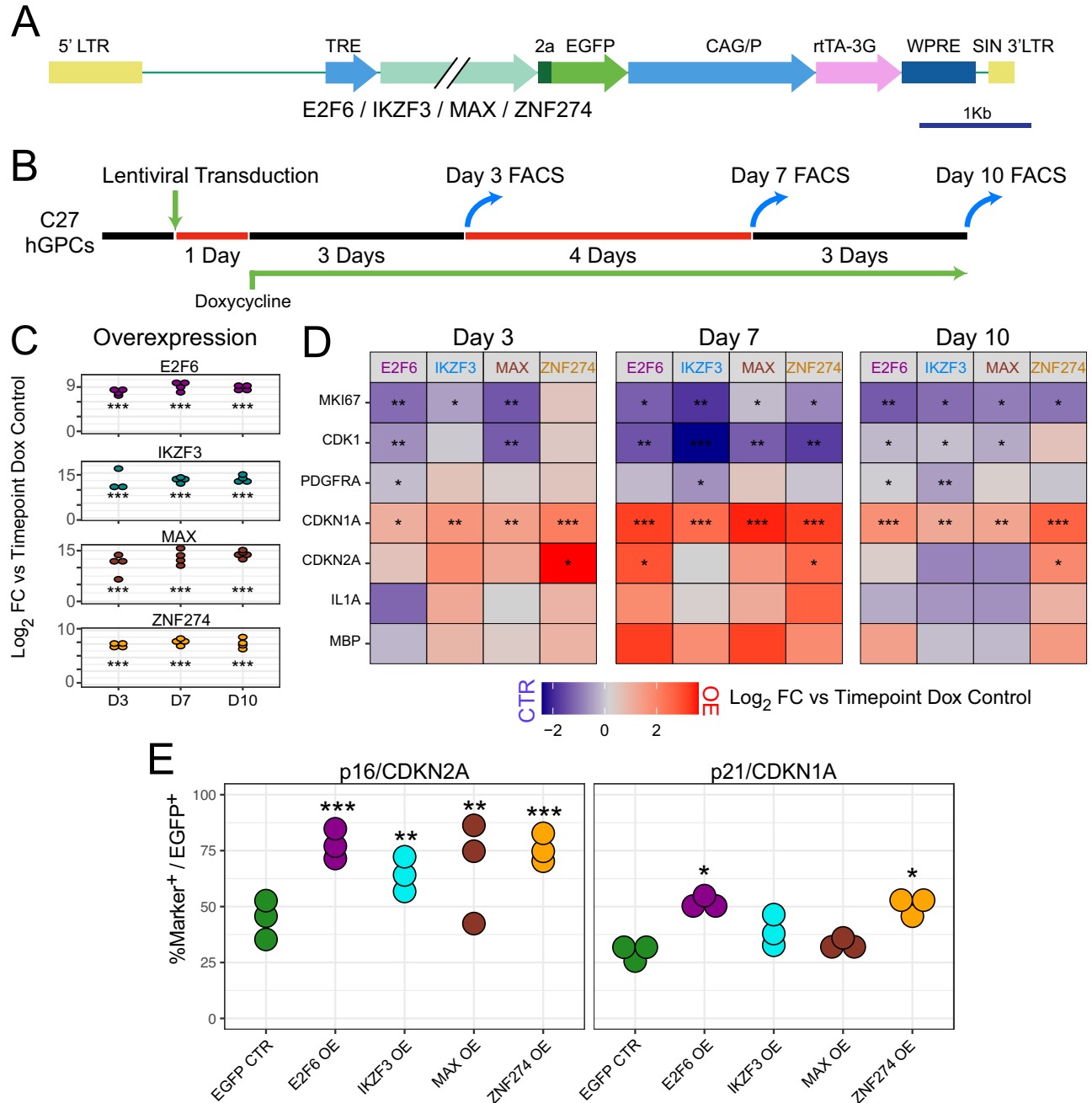

**Fig. 4 | Induction of features of aging via adult hGPC-enriched repressors.**
**A** Schematic outlining the structure of four distinct doxycycline (Dox)-inducible
EGFP lentiviral expression vectors, each encoding one of the transcriptional
repressors: E2F6, IKZF3, MAX, or ZNF274. **B** Induced pluripotent stem cell (iPSC)-
derived hGPC cultures (line C27[64,163]) were transduced with a single lentivirus or
vehicle for 1 day, and then treated with Dox for the remainder of the experiment. At
3, 7 and 10 days following initiation of Dox-induced transgene expression, hGPCs
were isolated via FACS for qPCR. **C** qPCRs of Dox-treated cells showing expression
of each transcription factor, vs matched timepoint controls ($n = 3$–5 independent

experiments/condition) **D** qPCR fold-change heatmap of select aging related genes.
Within timepoint comparisons to controls were calculated via post hoc estimated
marginal means tests of linear models following regression of a cell batch effect.
**E** Immunocytochemistry for p16 (left) and p21 (right) in EGFP+ cells at 7 days post
infection ($n = 3$ biologically independent samples). Post hoc pairwise comparisons
of each overexpression condition to control in (**C**–**E**) were calculated via estimated
marginal means tests of linear models, following regression of a cell batch effect.
FDR adjusted $p$-values: *<0.05, ** <0.01, *** <0.001. Source data are provided as a
Source Data file; exact p values are listed there. Error bars ± SEM.

of the age-associated changes in gene expression by otherwise young
hGPCs. To this end, we designed doxycycline (Dox) inducible over-
expression lentiviruses for each transcription factor (Fig. 4A). We first
identified which protein-coding isoform was most abundant in adult
GPCs for each repressor, so as to best mimic endogenous age-
associated upregulation; these candidates were E2F6-202, IKZF3-217,
MAX-201, and ZNF274-201 (Supplementary Fig. 3). These cDNAs were

cloned downstream of a tetracycline response element promoter, and
upstream of a T2A self-cleaving EGFP reporter (Fig. 4A). Human
induced pluripotent stem cell (iPSC)- or embryonic stem cell (ESC)-
derived hGPC cultures, were prepared as previously described[64], were
infected for 24 h, then treated with Dox to induce transgene over-
expression. C27 iPSC-derived or WA09 ESC-derived hGPCs (the C27
line was generously provided by L. Studer, and WA09 was purchased

from WiCell) were chosen as their transcriptomes closely resemble that of fetal GPCs (Supplementary Fig. 4A), and they are similarly capable of engrafting and myelinating dysmyelinated mice upon transplantation[64,65]. Transduced C27 hGPCs were selected via FACS for EGFP expression at 3, 7, and 10 days following Dox addition (Fig. 4B, $n = 3–5$). Uninfected cultures given Dox were used as controls. RNA was then extracted, and aging-associated genes of interest were analyzed by qPCR. Significant induction of each adult-enriched repressor was observed at each timepoint following Dox supplementation (Fig. 4C).

We found that upon induced heterochronic expression of these repressors, that both MKI67 and CDK1, genes whose upregulation are associated with active cell division, were significantly repressed by at least 2 of the 3 timepoints assessed (Fig. 4D). This was consistent with their diminished expression in adult hGPCs (Fig. 2F), and suggested their direct repression by E2F6, MAX, and ZNF274 (MKI67), or by all four (CDK1). The GPC stage transcript PDGFRA was also significantly repressed at one or two timepoints in both the IKZF3- and E2F6- transduced hGPCs, consistent with their repression in normal adult hGPCs. Interestingly, the senescence-associated cyclin-dependent kinase inhibitor CDKN1A/p21 was upregulated in response to each of the tested repressors at all timepoints, while CDKN2A/p16 was similarly upregulated at all timepoints in ZNF274-transduced hGPCs, and in the E2F6-over-expressing hGPCs at day 7 (Fig. 4D). In addition, the expression of both MBP and IL1A, each of which is strongly upregulated in adult hGPCs, trended up sharply in response to repressor transduction, although timepoint-associated variability prevented their increments from achieving statistical significance.

To further confirm the induction of aged signatures via overexpression of adult-enriched repressors, we stained for p16/CDKN2A and p21/CDKN1A in C27 hGPCs 7 days following dox addition. For this experiment we compared each overexpression condition to an EGFP CTR virus; this comparison confirmed that forced expression of each adult repressor induced significantly greater expression of p16/CDKN2A in EGFP⁺ cells; E2F6 and ZNF274 in particular were associated with higher expression of p21/CDKN1A as well (Fig. 4E). Interestingly, uninfected EGFP⁻ GPCs in the same cultures also exhibited higher p16/CDKN2A expression in response to E2F6, ZNF274 and MAX overexpression, suggesting the activation of a senescence-associated paracrine response in these cells (Supplementary Fig. 4B). Together, these data further supported our prediction that forced, premature expression in young hGPCs of the adult-enriched GPC repressors, ZNF274, E2F6, IKZF3 and MAX are individually sufficient to precociously trigger salient features of aging in human GPCs.

## GPC overexpression of E2F6 or ZNF274 drives age-associated transcriptional changes

To better understand the effects of these factors on hGPC transcriptional networks, we next used scRNA-seq (10X Genomics, v3.1) to assess the changes in gene expression associated with the two strongest factors—E2F6 and ZNF274—at their peak functional timepoint, 7 days post doxycycline. To do so, we transduced 160 DIV hGPC cultures with EGFP-tagged lentiviruses expressing either E2F6 or ZNF274, or an EGFP control, followed by FACS isolation on EGFP and single cell capture for scRNA-Seq. For differential expression, we narrowed the recovered populations to those specified as GPCs that also showed strong overexpression of their transduced factor (Fig. 5A, Supplementary Data 4). Compared to EGFP control GPCs, E2F6 overexpression yielded 584 differentially expressed genes, while ZNF274 overexpression was associated with the differential expression of 228 (Fig. 5B, Supplementary Data 4). Of 154 differentially expressed genes shared between the two, 152 were directionally concordant.

We next mined these genesets to validate the E2F6 and ZNF274 adult repressor targets identified from our in vivo bulk RNA-sequencing dataset (Fig. 3G, Supplementary Data 3). rCisTarget[26]

predicted numerous enriched motifs attributed to E2F6 or ZNF274 repression in their respective overexpression genesets with largely significant enrichment for predicted targets in repressed rather than activated genes (E2F6: $p < 2.2*10^{-16}$, ZNF274: $p = 4.5*10^{-11}$, Fisher's exact test, Supplementary Data 4). Of these predicted repressed targets that were also found to be repressed in adult GPCs, all of the E2F6 targets from our in vivo bulk analysis were recovered via our scRNA-seq analysis, along with an additional 33 repressed direct targets (Fig. 5C). In the case of ZNF274, our scRNA-seq analysis recovered all but 5 predicted direct targets from our in vivo bulk analysis, and added another 6 prospective direct targets (Fig. 5D). Next, we used AUCell[26] to score the activity of both regulons uncovered in our scRNA-Seq datasets of each group. This revealed the significantly repressed signatures of the E2F6 and ZNF274 regulons in their respective overexpression paradigms vs EGFP controls, as well as in each other's paradigm, due to their mutual repression of shared targets (E2F6: $p < 2.2*10^{-16}$; ZNF274: $p < 2.2*10^{-16}$; Wilcoxon rank sum test) (Fig. 5E).

Of these regulons, a cohort of genes associated with proliferation was sharply downregulated; these included TOP2A, PDGFRA, PCLAF, MKI67, HMGN2, HMGB2, HMGB1, CENPU, and CDK1 (Fig. 5G). At the same time, the senescence associated transcript CDKN1A/P21 was upregulated in both conditions, while genes typically absent or lowly-expressed in senescent cells, such as LMNB1 and TMPO/LAP2, were downregulated. Interestingly, fetal enriched transcription factors highlighted in our bulk RNA-seq analysis were also noted to be significantly downregulated; these included TEAD2, EZH2, and BCL11A (Fig. 5F, G). Among these curated genes, many were identified as direct targets of E2F6 and/or ZNF274 in addition to being similarly differentially expressed in primary GPCs during aging (Fig. 5H). Of those targeted by E2F6 and/or ZNF274 that were also differentially expressed in primary GPCs during aging, we validated via qPCR the repression of ETV1, NUSAP1, and PEG10 in both conditions and HMGB3 following E2F6 repression vs. an EGFP control virus (Fig. 5I).

To better define the network interactions leading to this expression pattern, we next analyzed both differentially expressed cohorts in IPA (Fig. 5J, Supplementary Data 4). This analysis yielded a strong signal of senescence across both overexpression paradigms, with significantly represented ontologies that included TP53, CDKN1A/P21, and CDKN2A/P16 signaling, mitochondrial dysfunction, DNA damage, and cell senescence per se. Among those downregulated ontologies were numerous terms tied to proliferation, including cell cycle progression, and those signaling pathways driven by YAP1, E2F3, PCLAF, TEAD1, CDK1, HMGA1, HMGB1, and PDGFRA.

We next assessed the mitotic index of these cells via immunolocalization of Ki67, the protein product of MKI67, in transcription factor-overexpressing hGPCs, vs both Dox-alone controls, and EGFP expressing CTR hGPCs (Fig. 5K). We found that both E2F6 and ZNF274 overexpression significantly reduced Ki67 incidence when compared to either the dox alone or EGFP control. Further, we assessed these cultures for the induction of beta-galactosidase activity, which is typically observed in aged cells. We found that hGPCs transduced to express E2F6 exhibited a significant upregulation of beta-galactosidase activity relative to both the EGFP-only virus and Dox-alone controls (Fig. 5L). Together, these data revealed downstream transcriptional and phenotypic changes in response to E2F6 or ZNF274 overexpression, which recapitulated salient features of aging by human GPCs, and which thereby supported our predicted network for the regulation of aging-associated transcription by human GPCs.

## The miRNA expression pattern of fetal hGPCs predicts their suppression of hGPC aging

To identify potential post-transcriptional regulators of gene expression, we assessed differences in miRNA expression between adult and fetal hGPCs ($n = 4$) utilizing Affymetrix GeneChip miRNA 3.0 arrays. PCA displayed segregation of both hGPC populations as defined by

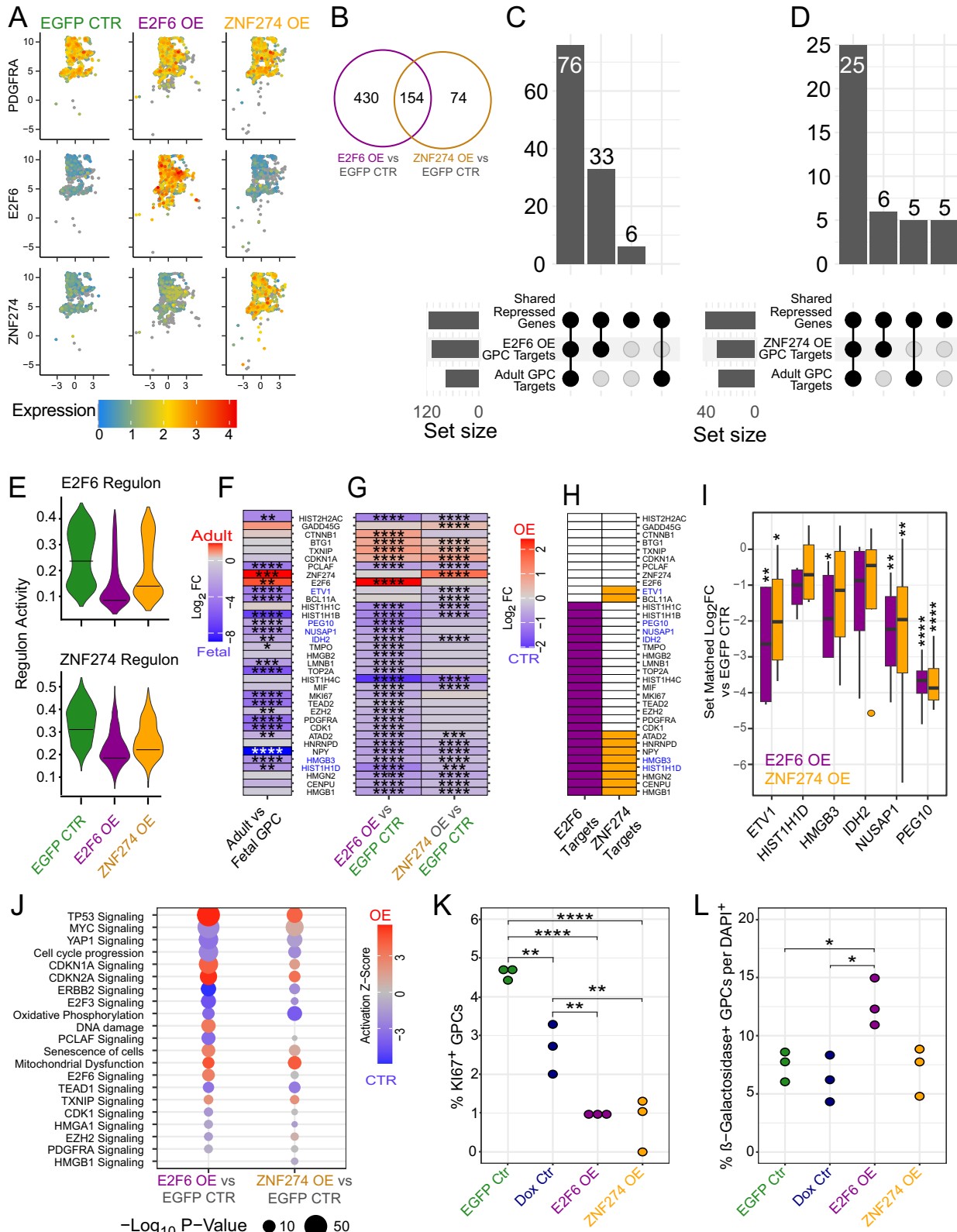

their miRNA expression profiles (Fig. 6A). Differential expression between both ages (adjusted *p*-value < 0.01) yielded 56 miRNAs (23 enriched in adult GPCs, 33 enriched in fetal GPCs, Fig. 6B, Supplementary Data 5). Notably among these differentially expressed miRNAs were the adult oligodendrocyte regulators miR-219a-3p and miR-338-5p[66,67] in addition to fetal progenitor stage miRNAs miR-9-3p, miR-9-5p[68], and miR-17-5p[69].

We next utilized this cohort of miRNAs to predict genes whose expression might be expected to be repressed via miRNA upregulation, separately analyzing both the adult and fetal hGPC pools. To accomplish this, we used miRNAtap to query five miRNA gene target databases: DIANA[70], Miranda[71], PicTar[72], TargetScan[73], and miRDB[74]. To maximize precision, genes were only considered a target if they appeared in at least two databases. Among fetal-enriched miRs, this

**Fig. 5 | E2F6 and ZNF274 overexpression each induce an aged transcriptome in hGPCs.** scRNA-sequencing of E2F6 and ZNF274 overexpression (OE) with a GFP lentivirus used as control, 7 days following doxycycline (Dox) treatment. **A** UMAPS of isolated GPC populations showing PDGFRA expression in addition to E2F6 and ZNF274 for demonstration of overexpression (EGFP CTR: 921 GPCs; E2F6 OE: 1085 GPCs, ZNF274 OE−906 GPCs). **B** Venn diagram of both differentially expressed genesets (FDR < 0.05, Log₂FC > 0.25). **C, D** Upset plots indicating repressed genes shared between fetal vs adult hGPCs and E2F6 I or ZNF274 (**D**) OE cells, and their respective targets, as determined via rCisTarget from these scRNA-seq OE experiments and the fetal vs adult bulk RNA-seq experiments. **E** Scoring via AUCell of the rCisTarget determined regulons of either E2F6 or ZNF274. Black lines indicate median expression. **F, G** Heatmaps indicating Log₂ fold changes and significances of adult vs. fetal GPCs (**F**) and E2F6 or ZNF274 vs EGFP CTR (**G**). **H** Significant direct targets of E2F6 or ZNF274 OE. **I** qPCR validation of select direct targets (indicated in blue in **F;** n = 4 biologically independent samples). Central lines indicate the median, boxes show the interquartile range, and the whiskers indicate 1.5 times the interquartile range. Outliers are plotted beyond the whiskers. **J** Dot plot of curated IPA terms (FDR < 0.05). **K** KI67 staining of EGFP⁺ cells (Dox Ctr is of DAPI⁺, n = 3 biologically independent samples). **L** β-Galactosidase⁺ staining of DAPI⁺ cells (n = 3 biologically independent samples). Statistics for **I, K,** and **L** were calculated via estimated marginal means tests of linear models, following regression of cell batch effect. FDR adjusted p-values: *<0.05, **<0.01, ***<0.001, ****<0.0001. Source data are provided as a Source Data file; exact p values are listed there. Error bars ± SEM.

approach predicted an average of 36.3 (SD = 24.5) repressed genes per miRNA. In contrast, among adult hGPC-enriched miRNAs, an average of 46.4 (SD = 37.8) genes were predicted as targets per miRNA (Fig. 6C, Supplementary Data 5). Altogether, this identified the potential repression of 48.8% of adult hGPC-enriched genes via fetal miRNAs, and repression of 39.9% of fetal hGPC-enriched genes by adult miRNAs.

To assess the functional importance of these miRNA-dependent post-transcriptional regulatory mechanisms, we curated fetal and adult networks according to miRNA targeting of functionally-relevant, differentially expressed genes (Fig. 6D, E). Our analysis predicted that the adult transcriptional regulators STAT3, E2F6, and MAX could be inhibited by a core set of 7 miRNAs in fetal hGPCs (Fig. 6D), 5 of which had already been shown to repress STAT3 in other cell types; these included miR-126b-5p, miR-106a-5p, miR-17-5p, miR-130a-3p, and miR-130b-3p[75–79]. In parallel, a number of early and mature oligodrocytic genes−including MBP, UGT8, CD9, PLP1, MYRF, and PMP22 - were concurrently targeted for inhibition, all consistent with maintenance of the progenitor state[80]. Importantly, a cohort of genes linked to either senescence, the inhibition of proliferation, or both, were also predicted to be actively repressed in fetal hGPCs. These genes included RUNX1, RUNX2, BIN1, DMTF1/DMP1, CTNNA1, SERPINE1, CDKN1C, PAK1, IFI16, EFEMP1, MAP3K7, AHR, OGT, CBX7, and CYLD[38,40–42,81–91]. Accordingly, the inhibition of senescence and/or activation of proliferation has been associated with a number of the fetal hGPC-enriched miRNAs that we identified, including miR-17-5p, miR-93-3p, miR-1260b, miR-106a-5p, miR-767-5p, miR-130a-3p, miR-9-3p, miR-9-5p, and miR-130b-3p[91–100]. Together, these data provide a miRNA-based mechanism complementary to direct transcription factor regulation, by which fetal hGPCs may maintain their characteristic progenitor transcriptional state and signature.

## Adult miRNA signaling may repress the proliferative progenitor state and augur senescence

We next inspected the potential miRNA regulatory network within adult hGPCs (Fig. 6E, Supplementary Data 5). Our analysis identified a cohort of 5 miRNAs whose expression was associated with the suppression of the fetal transcriptional regulators HDAC2, NFIB, BCL11A, TEAD2, and HMGA2. These miRNAs may operate in parallel to adult transcriptional repressors in inhibiting the expression of those genes involved in maintenance of the hGPC progenitor state; these fetal-enriched targets include PDGFRA, PTPRZ1, ZBTB18, SOX6, EGFR, and NRXN1. In addition, the adult miRNA environment was predicted to repress numerous genes known to induce a proliferative state or to delay senescence, including LMNB1[43], PATZ1[44], GADD45A[101], YAP1 and TEAD1[36], CDK1[102], TPX2[103], S1PR1[104], RRM2[105], CCND2[35], SGO1[106], MCM4 and MCM6[107], ZNF423[108], PHB[109], WLS[110], and ZMAT3[111]. In particular, adult hGPCs were enriched in a number of miRNAs already functionally linked to the induction of senescence or inhibition of proliferation; these included miR-584-5p[112], miR-193a-5p[113], miR-548ac[114], miR-23b-3p[115], miR-140-3p[116], and miR-330-3p[116]. The association of these miRs

with the age-dependent decline in hGPC proliferative competence, and the identification of their targets as developmental regulators of proliferation and expansion competence, suggests an important role for aging-associated miRNA networks, parallel to transcriptional repressors and cross-regulated by them, in the development by adult hGPCs of mitotic senescence.

## Transcription factor regulation of miRNAs consolidates hGPC identity

We next sought to predict the upstream regulation of differentially expressed miRNAs in fetal and adult GPCs by querying the TransmiR transcription factor miRNA regulation database[117]. This approach predicted regulation of 54 of 56 age-specific GPC miRNAs via 66 transcription factors that were similarly determined to be significantly differentially expressed between fetal and adult GPCs (Supplementary Fig. 5A, Supplementary Data 5). Interestingly, the top four predicted miRNA-regulating TFs were all MYC-associated factors, including MAX, MYC itself, E2F6, and the fetal-enriched MYC associated zinc finger protein, MAZ, targeting 36, 33, 30, and 28 unique differentially expressed miRNAs respectively.

Inspection of proposed relationships in the context of our 12 TF candidates (Fig. 3) identified a large number of fetal hGPC miRNAs as targets of both fetal activators and adult repressors; in contrast, those miRNAs enriched in adult GPCs were more uniquely targeted (Supplementary Fig. 5B). In fetal hGPCs, MYC was predicted to drive the expression of numerous miRNAs, that were predicted to be repressed in turn in adulthood, via E2F6, MAX or both. miR-130a-3p in particular was predicted to be targeted by MYC, MAX, and E2F6, in addition to activation via TEAD2. Among validated TF-miRNA interactions in other cell types, upregulation of fetal miR-17-5p by MYC, and its repression by MAX[118–121], has been reported. Similarly, both MYC and TEAD2/YAP1 can activate miR-130-3p[96,97,122], which is high in fetal hGPCs, and MYC can activate miR-9[123], which falls with oligodendrocytic maturity[68]. Indeed, oligodendrocyte maturation was attended by the upregulation of miR-219a-2-3p, which may be inhibited in fetal hGPCs via EZH2.

In adult hGPCs, unlike fetal, only two miRNAs, miR-151a-5p and miR-4687-3p, were predicted to be regulated by both an adult activator and fetal repressor, while one miRNA, miR-1268b, was predicted to be inhibited by both EZH2 and HDAC2 in parallel. Interestingly, STAT3, whose increased activity has been linked to senescence[124], was predicted to drive the expression of a set of miRNAs independently associated with the induction of senescence; these included miR-584-5p, miR-330-3p, miR-23b-3p, and miR-140-3p.

We also inspected the differentially expressed genesets from our E2F6 and ZNF274 overexpression studies (Fig. 5), so as to identify any miRNAs which might further contribute to the control of gene expression by aging GPCs. Among those genes downregulated in response to E2F6 or ZNF274 overexpression, none were predicted targets of adult-enriched miRNAs (Supplementary Data 5). In contrast, among those genes upregulated in either E2F6- or ZNF274-transduced GPCs, a number were found to be targeted by fetal GPC miRNAs

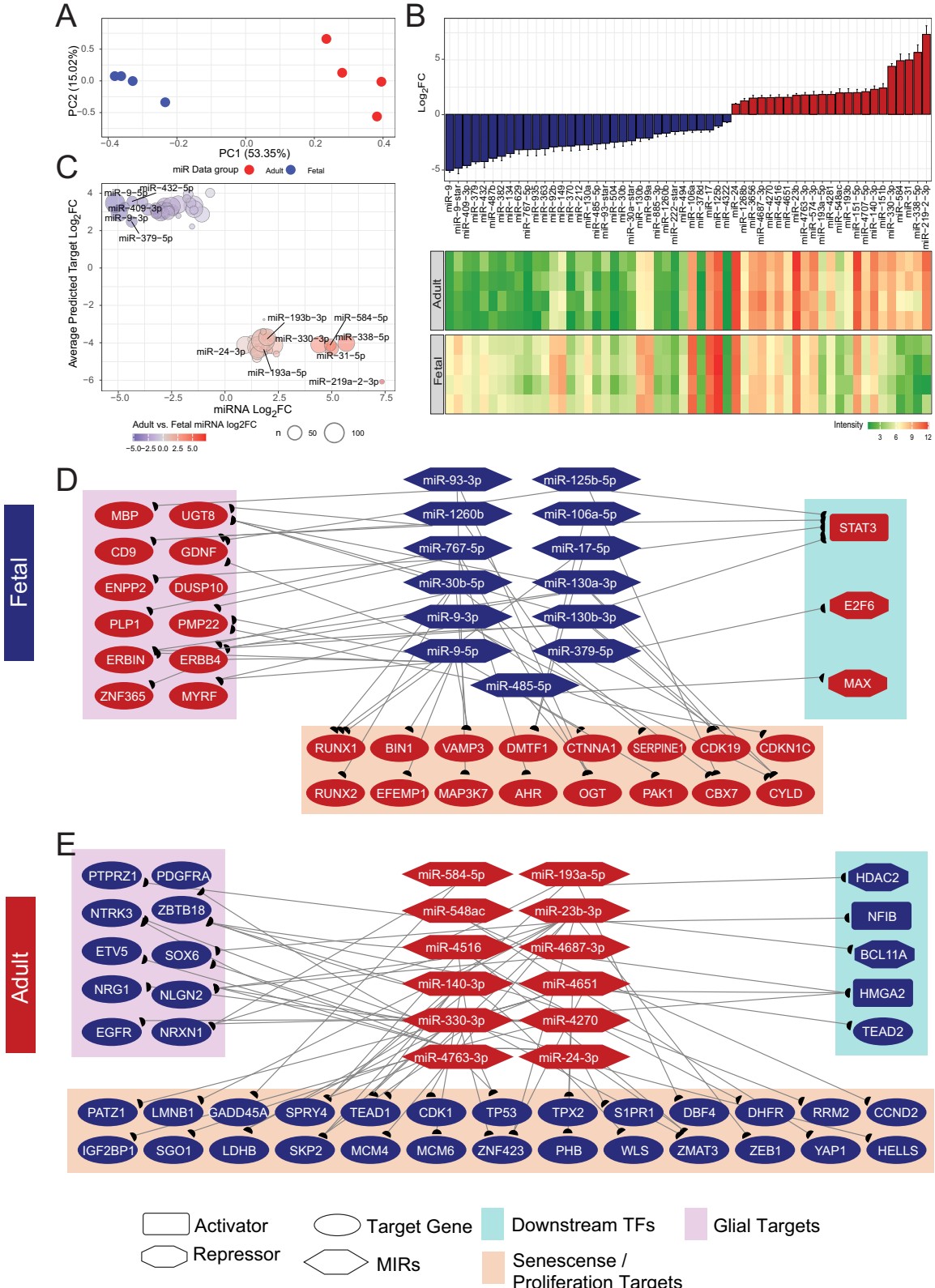

**Fig. 6 | MicroRNAs drive hGPC transcriptional maturation in tandem with transcription factor activity. A**. Principal component analysis of miRNA microarray samples from human A2B5⁺ adult and CD140a⁺ fetal GPCs (*n* = 4 biologically independent samples). **B**. Log2 fold change bar plots and heatmap of all differentially expressed miRNAs (FDR < 0.01, calculated in limma). **C** Characterization bubble plot of enrichment of miRNAs, versus the average log2 FC of its predicted gene targets. **D**, **E** Curated signaling networks of (**D**) fetal (*top*) and (**E**) adult (*bottom*) enriched miRNAs and their predicted targets. Source data are provided as a Source Data file. Error bars ± SEM.

(Supplementary Fig. 5C). These data suggest that the expression of aging-associated, adult GPC gene sets may, at least in part, be facilitated via the E2F6 or ZNF274 mediated repression of fetal miRNAs that might otherwise suppress the expression of these aging-associated transcripts. Of particular note in this regard, E2F6 was predicted to repress a group of these fetal-enriched GPC miRNAs in our transmiR analysis (Supplementary Fig. 5B); these included miR-106a-5p, miR-130a-3p, miR-130b-3p, miR-92b-3p, miR17-5p, miR-363-3p, and miR-504-5p. Through integration of transcriptional and miRNA profiling, pathway analyses, and target predictions, these data suggest a model of human GPC aging, whereby fetal hGPCs maintain progenitor gene expression and activate proliferative programs, while repressing oligodendrocytic and senescent gene programs, both transcriptionally and post-transcriptionally via microRNA. With adult maturation and the passage of time as well as population doublings, hGPCs begin to upregulate repressors of these youthful networks, while activating programs to advance a progressively more differentiated, and ultimately senescent, phenotype.

### Glial Explorer: an interactive database to query human glial gene expression

To enable independent exploration of these and related data, we have developed a publicly-available application (accessible at http://GlialExplorer.io/), that comprises our single cell RNA-Sequencing datasets for human glial progenitor cells and their derivatives[125]. In addition, the accompanying Shiny app[126] (under Bulk Data; Supplementary Fig. 6A) provide bulk RNA-seq data for exploration, as covered both here and in previous studies[65,127,128]. Briefly, the scRNA-Seq data page can be explored by both feature and violin plot (Supplementary Fig. 6B). The bulk data page allows simple querying of gene abundances across datasets (Supplementary Fig. 6C), as well as the incidence of predicted splice variants (Supplementary Fig. 6D). Our intention is that this app and its included database should enable interested researchers to quickly survey their genes of interest, and to interactively assess their regulation in human glial ontogeny and aging.

## Discussion

Human glial progenitors first appear in the 2nd trimester of human development, after which a parenchymal pool remains throughout life[25,129]. In early development and youth, these progenitors are highly proliferative and self-renewing. Yet their ability to divide and replenish lost myelin decreases substantially with age, as well as in the setting of antecedent demyelination and white matter disease[13,15,130–139]. Given the evolutionary divergence and marked anatomic and functional differences between murine and human glia[140], we focused on human glia when assessing the basis for this loss of expansion potential, so as to efficiently identify therapeutically relevant targets. As such, we used bulk RNA-sequencing of freshly isolated fetal and adult human GPCs, together with single cell RNA-sequencing of fetal hGPCs, to track divergent transcriptional changes in our population of interest, while minimizing off-target cell-type contaminants. Bipotential oligodendrocyte-astrocyte glial progenitor cells were best identified by PDGFαR/ CD140a expression in fetal development; these cells were largely captured by the A2B5⁺/PSA-NCAM⁻ phenotype as well. In contrast, analogous glial progenitor cells were best identified in adult brain by the A2B5 phenotype alone, as adult GPCs largely downregulated PDGFαR expression with maturation, despite retaining the functional characteristics of GPCs. Our approach recognized that the transcriptional hallmarks of hGPC identity may evolve with maturation and aging, such that the nominally canonical markers of the GPC phenotype at one point in differentiation may not be so at an earlier of later stage. This strategy provided us with a set of genes whose expression distinguished fetal hGPCs from their aged successors, and which suggested a progressive bias with aging towards mitotic suppression and terminal differentiation. This observation is in accord with previous rodent GPC gene expression and proteome data, which noted the downregulation with aging of progenitor markers such as CSPG4, PDGFRA, and PTPRZ1, pari passu with the upregulation of early oligodendrocyte markers such as MBP, CNP, and MOG[14,16]. Importantly, we found that adult human GPCs similarly acquire an expression signature consistent with a loss of proliferative competence, coupled with an upregulation of an ensemble of senescence-linked genes.

Our analysis predicted that MYC, whose expression was enriched in fetal hGPCs, is a central regulator of proliferative capacity of human GPCs, through its transcriptional regulation of a set of downstream genes and miRNAs that coordinately and positively regulate mitotic competence and cell cyclicity. MYC has previously been identified as an important modulator of both the epigenetic landscape and proliferation of murine GPCs, via the activation of CDK1[141]. Moreover, MYC has recently been extensively studied as mitogenic for adult murine GPCs and an inhibitor of their senescence[37], functions consistent with the MYC-regulated targets of the repressive network that we have identified in human GPCs. Furthermore, MYC also may convey a competitive advantage to human GPCs that declines with age[125]. Indeed, our model suggests the direct repression in adult hGPCs not only of MYC, but also of many of its targets as well. Interestingly, IKZF3 has been reported to directly suppress MYC in pre-B-cells, limiting their proliferative ability[60]. For its part, MAX can complex with MYC to both inhibit its function, and to alter its transcriptional targets. Furthermore, MAX and E2F6 can both target MYC binding sites competitively, in addition to the E2F sites that E2F6 typically represses[63]. MYC's downregulation has also been reported to follow the upstream activation of AHR[49] and BIN1[81], each of which was upregulated in our adult GPC dataset. Interestingly, adult hGPCs were also enriched for ZNF274, whose predicted presence at MYC targets may attract a complex of TRIM28/KAP1, SETDB1, and ATRX to induce heterochromatization of these genes[142]. MYC was also predicted to activate an ensemble of miRNAs in fetal GPCs, many of which were predicted to be counter-regulated by E2F6 and MAX in adult GPCs. Among these were the MYC-regulated miRNAs miR-9 and miR-130a-3p, each of which has previously been linked to delayed senescence[79,98,123].

Interestingly, miR-130a-3p was also predicted to repress another highly active adult hGPC transcriptional activator, STAT3, whose expression is necessary for glial development[143], remyelination[144], and has been implicated as a driver of senescence[75,79,145]. Indeed, miRNA-130a-3p repression of STAT3 delays senescence in renal tubular epithelial cells, as driven by metformin[79], a drug similarly shown to enhance remyelination by aged rat GPCs[16]. Similarly, STAT3 expression may increase in hGPCs after exposure to conditioned media taken from cultures of iPSC-derived neural progenitor cells, generated from patients with primary progressive multiple sclerosis[139]. Of note, our analysis also predicted STAT3 activation of miRNAs that included miR-23b-3p, the most highly upregulated miR in senescent mesenchymal cells[92].

Further assessment of our miR differential expression data revealed a number of post-transcriptional regulatory mechanisms poised to modulate fetal and adult hGPC transcription. This included the upregulation in adult hGPCs of well-studied regulators of oligodendrocyte maturation, miR-219 and miR-338[66,67], consistent with the more mature oligodendrocytic transcriptional signature of adult GPCs. In that regard, the adult GPC-enriched miRNAs miR-338-5p, miR-219a-2-3p, and miR-584-5p, have all previously been reported to be among the most highly upregulated miRs in the white matter of multiple sclerosis (MS) patients, compared to healthy controls[146]. Accordingly, miR-130a-3p, miR-9-3p and miR-9-5p, each of which has been found to be downregulated in MS white matter, were also downregulated in our adult hGPCs. These findings are consistent with the hypothesis that recurrent progenitor mobilization in relapsing-remitting MS mimics transcriptional events that accrue with aging. Additional miRNAs,

including miR-17-5p and miR-93-3p were also predicted by our analysis to participate in maintaining the progenitor state of fetal hGPCs, while miR-584-5p, miR-330-3p, miR-23b-3p, and miR-140-3p were predicted to promote senescence in adult hGPCs.

The heterogeneity of adult hGPCs has been postulated to increase in the adult brain in a region-specific manner[13]. As such, future studies incorporating scRNA-sequencing from multiple regions, paired with spatial transcriptomics, will be needed to better understand the regional geography of normal glial aging, and its relationships with neuronal activity and vascular health[147,148]. The transcriptional correlates to glial aging in the setting of disease, both neurodegenerative and dysmyelinating, will then be needed to assess the interaction of pathology with normal aging, as well as the response of aging cells to the broad variety of disease processes to which they may be exposed. In this regard, it will be critical to account for the effects of non-cell autonomous drivers of GPC aging, such as diminished local vascular perfusion and astrocytic support[149], on glial aging and senescence. Taken together, given the distinctions between young and aged hGPCs, and the extent to which their transcriptomes can be regulated via the mechanisms we have described, it may be feasible to safely rejuvenate aged human GPCs to a more expansion-competent and phenotypically malleable phenotype, enabling them to more effectively compensate for the ill effects of aging and adult white matter disease.

## Methods

### Ethics
Human brain samples were obtained, and all experiments performed, under approved protocols of the Institutional Review Board of the University of Rochester Medical Center. Adult samples were acquired under protocol study 00000150, and fetal samples were obtained as deidentified tissue, classified as exempt. Samples were obtained for research use from consenting patients or their guardians at Strong Memorial Hospital at the University of Rochester.

### Brain tissue cell isolation
Brain tissue was obtained from normal GW 18–22 cortical and/or VZ/SVZ dissections or adult white matter/cortex epileptic resections (18 F, 19 M, and 27 F years old for mRNA, 8 M, 20 F, 43 M, and 54 F years old for miRNA). **Adult** tissue acquisition, dissociation and immunomagnetic sorting of A2B5+ cells for adult hGPCs was performed as described[12]. **Fetal** GPCs were isolated from dissociated tissue via CD140a-directed MACS or FACS, on a BD FACS AriaIIIU[22]. Fetal PSA-NCAM-/A2B5+ hGPCs were isolated via FACS by first staining for PSA-NCAM in Miltenyi washing buffer (MWB: PBS, 0.5% BSA Fraction V (ThermoFisher cat. no. 15260037)), 2 μM EDTA (ThermoFisher cat. no. 15575020), a 10 min wash in MWB (200 g), a 20 min incubation of APC conjugated secondary antibody, another 10 min wash (200 g), a 20 minincubation with PE conjugated A2B5 (clone 105, ATCC, Manassas, VA), and a final wash (200 g). In addition to antigenic selection above, samples were also depleted of DAPI+ cells during FACS (Supplementary Fig. 7).

### Bulk RNA-sequencing
RNA was purified from isolates via Qiagen RNeasy kits and bulk RNA sequencing libraries were constructed. Samples were sequenced deeply on an Illumina HiSeq 2500 at the University of Rochester Genomics Research Center. Raw FASTQ files were trimmed and adapters removed using fastp[150] and aligned to GRCh38 using Ensembl 95 gene annotations via STAR in 2-pass mode across all samples[151] and quantified with RSEM version 1.3.1[152]. Subsequent analysis was carried out in R[153] where RSEM gene level results were imported via tximport[154]. DE analysis was carried out in DESeq2[155] on all genes where paired analyses (Fetal A2B5+ vs CD140a+, fetal CD140a+ vs CD140a-) had paired information added to their models. For adult vs fetal DE analysis, age was concatenated with sort marker (CD140a- samples were not included) to define the group variable where sequencing batch was also added to the model to account for technical variability. Genes with an adjusted p-value < 0.01 and an absolute log2-fold change > 1 were deemed significant. These data were then further filtered by meaningful abundance, defined as a median TPM (calculated via RSEM) of 1 in at least 1 group (20,663 genes met this criterion prior to DE) for downstream analyses (IPA, Networks).

### scRNA-Seq analysis of fetal brain
The fetal brain sample as processed as above for bulk RNA-seq, until single cells were sorted via FACS for either CD140a+ or PSA-NCAM-/A2B5+ surface expression. Single cells were then captured on a 10X genomics Chromium controller utilizing v2 chemistry and libraries generated according to the manufacturer's instructions. Samples were sequenced on an Illumina HISeq 2500 system. Demultiplexed samples were then aligned and quantified using Cell Ranger to an index generated from GRCh38 and Ensembl 95 gene annotations using only protein coding, lncRNA, or miRNA biotypes. Analysis of scRNA-Seq samples was carried out via Seurat[24] within R. Both samples were merged and low-quality cells filtered out as defined by having mitochondrial gene expression > 15% or having < 500 unique genes. Samples were then normalized utilizing SCTransform taking care to regress out contributions due to total number of UMIs, percent mitochondrial gene content or the difference in S phase and G2M phase scores of each cell. PCA was then calculated, UMAP was run using the first 30 dimensions with n.neighbors = 60 and repulsion.strength = 0.8. FindNeighbors was then run followed by FindClusters with resolution set to 0.35. Based on expression profiles of each cluster, some similar clusters were merged into broader cell type clusters. Static differential expression of clusters was computed using the MAST test[156] where an adjusted p-value of < 0.01 and an absolute log2 fold change of > 0.5 was deemed significant. Prediction of active transcription factor regulons was carried out with the SCENIC package in R[26] using the standard workflow and the hg38 databases located at https://resources.aertslab.org/cistarget/. Genes were included in co-expression analyses if they were expressed in at least 1% of cells before the GENIE3 step.

### Ingenuity pathway analysis and network construction
Differentially expressed genes were fed into Ingenuity Pathway Analysis (Qiagen) to determine significant canonical, functional, and upstream signaling terms. For construction of the IPA network, terms were filtered for adjusted p-values below 0.001. Non-relevant IPA terms were removed along with highly redundant functional terms assessed via jaccard similarity indices using the iGraph package[157]. Modularity was established within Gephi[158] and the final network was visualized using Cytoscape[159]. Genes and terms of interest were retained for visualization purposes. Modules were broken out from one another and organized using the yFiles organic layout.

### Inference of transcription factor activity
Adult and fetal enriched gene lists were fed separately into RcisTarget[26] to identify overrepresentation of motifs in windows around the genes' promoters (500 bp up/100 bp down and 10 kb up and 10 kb down). Transcription factors that were associated with significantly enriched motifs (NES > 3) were then filtered by their significant differential expression in the input gene list. Within each window and gene list, only appropriate TF-gene interactions (Repressors downregulating genes and activators upregulating genes) were kept. Scanning windows were then merged to produce TF-gene edge lists of predicted fetal/adult repressors/activators. We finally narrowed our TFs of interest to those primarily reported as solely activators or repressors in the literature. For overexpression transcription factor activity analysis, repressed gene lists were used within RcisTarget and enrichment of these genesets were calculated using AUCell[26].

## miRNA microarray analysis

Adult (A2B5+; n = 4) and fetal (CD140a+; n = 4) hGPC cell suspensions were isolated via MACS as noted above, and their miRNA isolated using miRNeasy kits according to manufacturer instructions (QIAGEN). Purified miRNA was then prepared and profiled on Affymetrix Gene-Chip miRNA 3.0 Arrays as instructed by their standard protocol. Raw CEL files were then read into R via the oligo[160] package and samples were normalized via robust multi-array averaging (RMA). Probes were then filtered for only human miRNAs according to Affymetrix's annotation, and differential expression was carried out in limma[161] where significance was established at an adjusted $p$-value < 0.01. Finally, differentially expressed miRNAs were surveyed across five independent miRNA prediction databases using miRNAtap[162] with min_src set to 2 and method set to "geom". Transcription factor regulation of miRNAs was carried out via querying the TrasmiR V2.0 database[117].

## Exploratory analysis and visualization

PCA of bulk RNA-Seq or microarray samples was computed via prcomp with default settings on variance stabilized values of DESeq2 objects. PCAs were plotted via autoplot in the ggfortify package. Volcano plots were generated using EnhancedVolcano. Graphs were further edited or generated anew using ggplot2 and aligned using patchwork.

## Production of GPCs from human iPSCs and ESCs

Human induced pluripotent stem cells (C27[163]) or embryonic stem cells (WA09) were differentiated into GPCs using our previously described protocol[64,65,127]. Briefly, cells were first differentiated to neuroepithelial cells, then to pre-GPCs, and finally to GPCs. GPCs were maintained in glial media supplemented with T3, NT3, IGF1, and PDGF-AA, and typically harvested for use at 160 ± 10 DIV, with further CD140a/PDGFαR-based FACS purification of hGPCs when noted.

## Lentiviral expression of candidate transgenes

For overexpression of E2F6, ZNF274, IKZF3, or MAX, we first identified the most abundant protein coding transcript of each of these genes from the adult hGPC dataset. cDNAs for each transcript were cloned downstream of the tetracycline response element promoter in the pTANK-TRE-EGFP-CAG-rtTA3G-WPRE vector. Viral particles pseudotyped with vesicular stomatitis virus G glycoprotein were produced by transient transfection of HEK293FT cells and concentrated by ultracentrifugation, and titrated by QPCR (qPCR Lentivirus Titer Kit, ABM-Applied Biological Materials Inc). iPSC (C27) derived GPC cultures (160-180 days in vitro) were infected at 1.0 MOI in glial media for 24 h. Cells were washed with HBSS and maintained in glial media supplemented with 1 μg/ml doxycycline (Millipore-Sigma St. Louis, MO) for the remainder of the experiment. Transduced hGPCs were isolated via FACS on DAPI-/EGFP+ expression 3, 7, and 10 days following the initial addition of doxycycline. Doxycycline control cells were sorted on DAPI- alone.

## Quantitative PCR

RNA from overexpression experiments was extracted using RNeasy micro kits (Qiagen, Germany). First-strand cDNA was synthesized using TaqMan Reverse Transcription Reagents (Applied Biosystems, USA). qPCR reactions were run in triplicate by loading 1 ng of RNA mixed with FastStart Universal SybrGreen Mastermix (Roche Diagnostics, Germany) per reaction and analyzed on a real-time PCR instrument (CFX Connect Real-Time System thermocycler; Bio-Rad, or QuantStudio 12 K Flex system, Thermo Fisher). Results were normalized to the expression of 18 S from each sample. All primer sequences are provided in Supplementary Data 6.

## Immunocytochemistry

For immunocytochemistry, cells were fixed with 4% PFA for 15 min at 25 °C at 7 days after starting doxycycline treatment, then washed with PBS and permeabilized and blocked for 1 h in staining buffer (PBS + 1% BSA, 0.3% Triton X-100). The cells were then incubated with chicken anti-GFP (Aves GFP-1020), mouse anti-CDKN2A/p16INK4a (abcam ab54210), rat anti-MKI67 (Invitrogen 14-5698-82), or rabbit anti-CDKN1A/p21 (abcam ab109520). After washing with PBS, they were stained with secondary antibodies and DAPI for 30 min. Immunostained cultures were imaged on the ImageXpress Confocal HT.ai (Molecular Devices) at 10x magnification, sampling from 9 tiles spaced across each well. Images were segmented and quantified using MetaXpress version 6.7.0.211 (Molecular Devices). Beta-galactosidase was stained using a Senescence beta-Galactosidase Staining Kit according to manufacturer's instructions (Cell Signaling Tech., 9860 S). Cells were counterstained with DAPI, and 3 images spaced across each well were quantified. All antibody information, including targets and clones, source and catalog numbers as well as dilutions, is provided in Supplementary Data 7.

## scRNA-seq analysis of transduced hGPCs

For analysis of E2F6 and ZNF274 overexpression scRNA-seq experiments, an EGFP virus was used as a control[27]. hGPCs were differentiated from iPSCs (C27 line) and isolated via FACS as described above, then captured on a 10X chromium controller (V3.1 chemistry), and libraries were generated according to manufacturer's instructions. Samples were sequenced on an Illumina NovaSeq 6000 system at the Genomics Research Center at the University of Rochester. Demultiplexed samples were then aligned and quantified using StarSolo to an index generated from GRCh38 and Ensembl 95 gene annotations, using only protein coding, lncRNA, or miRNA biotypes with the eGFP transcript added[164]. Analysis of scRNA-Seq samples was carried out via Seurat[24] within R. Both samples were merged, and low-quality cells filtered out as defined by mitochondrial gene expression > 15%, or having < 500 unique genes. Samples were independently normalized using SCTransform with mitochondrial percentage, UMI counts, and the difference in computed cell cycle scores regressed. 3000 features were used for integration in Seurat with eGFP, ZNF274, and E2F6 removed from consideration for this step. Clusters were found at a resolution of 0.25 with 50 dimensions considered. This yielded 8 clusters, which were simplified to four major cell types, that included GPCs, NPCs, neuronal cells and astrocytic cells as determined manually by gene expression. For differential expression, only those genes expressed by at least 10% of any sample's GPCs were considered. The MAST test was used with Seurat (Log2 FC > 0.25, adjusted $p$ < 0.05), with E2F6 or ZNF274 overexpressing cells compared to their EGFP-only controls in the GPC population. Bulk and scRNA-sequencing data may be explored in our Shiny app at http://GlialExplorer.io/.

## Quantification and statistical analysis

For qPCR experiments of all four adult repressors, significant differences in delta CTs for each gene were analyzed in linear models constructed by the interaction of overexpression condition and timepoint with the addition of a cell batch covariate. Post hoc pairwise comparisons were tested via estimated marginal means tests against the Dox control within timepoints using the emmeans package[165]. qPCR experiments for validation of E2F6 or ZNF274 direct targets followed a similar strategy of linear model construction of both conditions interacting with genes of interest and a cell batch effect with pairwise comparison derived the same way. Linear models and pairwise comparisons of ICC and β-Galactosidase were evaluated similarly. Enrichment of genesets were tested via Fisher Exact Tests. P-values were adjusted for multiple comparisons using the false discovery rate method whereby $p$-values < 0.05 were deemed significant.

# Materials availability

Cells and reagents generated in this study will be made available on request. All requests for materials and reagents should be directed to

the Lead Contact (steven_goldman@urmc.rochester.edu). For some cellular reagents and plasmids, a completed Material Transfer Agreement may also be required.

## Data availability

Source data are provided with this paper. The gene expression data in this study have been deposited in GEO under accession numbers GSE165996 and GSE255589. Data were aligned to the GRCh38 genome and annotated using Ensembl v.95. All of our expression data are also available and may be further interrogated at GlialExplorer.io. TransmiR data from https://www.cuilab.cn/files/images/transmir2/download/literature/hsa.tsv.gz were obtained, and SCENIC/Rcistarget rankings were downloaded from https://resources.aertslab.org/cistarget/. Source data are provided with this paper.

## Code availability

All R Scripts may be accessed at https://github.com/CTNGoldmanLab/Human_GPC_Aging_2021. Further requests for information should be directed to and will be fulfilled by the Lead Contact (steven_goldman@urmc.rochester.edu).

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

## Acknowledgements

This work was supported by the Adelson Medical Research Foundation, and NIH grants R01NS110776 and R01AG072298 to S.G, as well as by

CNS2, Inc., and by a Novo Nordisk Foundation grant to P.M. We thank Roman Luštrik (Genialis) for assistance in deploying the Shiny app, and Lorenz Studer (Memorial Sloan-Kettering) for his generous provision of the C27 hiPS cells.

## Author contributions

D.C.M., N.K. and J.N.M. prepared and sorted the primary cells used for the experiments; N.P.T.H. and J.N.M. processed and analyzed the genomic data; E.R.K., N.P.T.H. and J.N.M. designed the shiny app; A.B. and D.K. generated the overexpression lentiviruses; P.M.M., C.P. and J.B. prepared the human iPSC-derived GPCs used for lentiviral overexpression experiments; B.M. and D.S. generated the quantitative PCR data; B.M. and D.S. prepared and analyzed the ICC data; J.N.M. and S.A.G. designed the study and wrote the paper. All authors approved the final manuscript.

## Competing interests

Dr. Goldman is also a part-time employee and stockholder of Sana Biotechnology (Seattle, WA), and his lab receives sponsored research support for projects unrelated to the present work from Sana. Dr. Goldman is also a stockholder and SAB member of CNS2, Inc.; his lab also receives research support from CNS2. N.P.T.H is now a Sana employee, and D.C.M. is a consultant to Sana, in both cases for unrelated work. None of the other authors have any known potential conflicts of interest with regards to this work.
