## [Peer Review File · Nature Communications]

Repression of developmental transcription factor networks triggers aging-associated gene expression in human glial progenitor cellsREVIEWER COMMENTS

Reviewer #1 (Remarks to the Author):

In this manuscript the authors use RNA sequencing to characterize the changes that occur in oligodendrocyte/glial precursor cells with age. Through a combination of cell specific bulk and single cell sequencing they show that adult human glial precursors have significantly distinct profiles from their fetal counterparts. Analysis of the differences between fetal and adult purified cell population revealed a surprisingly larger number of transcriptional differences. Pathway and network analyses identify regulators of MYC signaling as a central element and overexpression of transcriptional repressors alters relevant gene expression in iPSC derived cell lines. A number of miRNAs are also identified that may contribute to regulation of a mature glial phenotype.

This is an interesting paper that provide new insights into the maturation of oligodendrocytes in the human CNS. Previous studies have suggested that signaling through the MYC pathway is an important regulator of OPC aging and the current manuscript contains significant new data that extends and support this hypothesis. While the central take-home message of the manuscript is relatively straight forward, the current paper includes a significant amount of data and studies that are not directly relevant to control of OPC/GPC gaining. This includes much of the data in Figure 1 and 2. Indeed the A2B5/CD140a comparators are somewhat confusing and not further explored in the manuscript. A somewhat shortened and more focused manuscript would be easier to follow and potentially more impactful.

One general concern with the current paper is that while there is a lot of transcriptional analyses that implicate both a repressor network and the role of a number of miRNAs in the aging of glial precursors, there is very limited experimental data to demonstrate the functionality of the proposed pathways. Such validation may be beyond the scope of the current paper, but the authors should temper their statements in its absence.

Some other minor issues:

1. Related to the comment above, in Figure 5 the authors show the effects of expressing transcriptional repressors in iPSC derived hGPCs. The data suggests the induction of an aged transcriptome but make no comment on the proliferative or morphological characteristics of the cells. Specifically, it would be interesting to know if there was a reduction in cell division and an increased tendency to differentiate in the treated cells.

2. The authors should attempt to reduce the degree of abbreviations throughout.

Examples include defining the different cell types discussed on page 4. What is a pre-GPC? It is never articulated.

Overall, this is an interesting and well-done paper that will be of significant interest to workers in the field. Some refinement of the current manuscript would likely enhance its impact.

Reviewer #2 (Remarks to the Author):

Mariani et al provide a study aimed at understanding why there is reduced re-myelination with aging taking advantage of their access to primary human OL lineage cells and OL lineage cells derived from ipscs. The main message is derived by comparisons of the transcriptomes of CD140a+ (PDGFR +) cells derived from 17-20 week human fetal material and A2B5 + cells derived from adult human brain (ages 18-27). They identify age related transcriptional changes that repress genes involved in mitotic expansion and provide validation of the contribution of these genes by assessing effects of their overexpression in GPCs derived from an ipsc line. Establishing that these regulatory pathways contribute to age related myelination failure would be a significant contribution.

Overall challenges to answering why there is inadequate remyelination in acquired human disease include what is the OL lineage cell most responsible for such a process and how is it impacted by aging. The comparative transcriptome focused studies involve CD140a+ (PDGFR +) cells derived from 17-20 week human fetal material and A2B5 + cells derived from adult human brain (ages 18-27). Thus these comparisons involve cells at different stages of the OL lineage rather than the same cell young versus older adults. The materials used in the current study do not allow for more matched comparisons.

The authors show that CD140a+ is down regulated on the adult A2B5+ progenitor cells isolated from the human adult brain specimens and that no CD140a+ population is derived from this material. In situ analyses of human brain including by the current authors (Osorio GLIA 2023) and the Allen brain atlas indicate that CD140a+/NG2 cells are abundant in adult human brain and thus the issue whether current techniques are inadequate to recover such cells. Perhaps tissue based analytic techniques will be needed. The converse of isolating equivalent A2B5+ cells is also problematic as A2B5+ cells in the fetus are not yet all OL lineage committed; the adult A2B5+ cells express later lineage markers including MBP and are not present in the fetal material. If the primary aim of the study is to assess proliferation and senescence related mechanisms as a function of aging, it would seem more informative if one compared younger and older adults as done in rat related studies such as done by Neumann et al using A2B5+ cells. Late lineage cells are rarely present in fetal material (equivalent of adult A2B5+ cells).

The bulk and scRNAseq data of the manuscript are analyzed using standard tools and approaches. Specifically, the authors have taken into account the effect of different batches and included this in their differential gene expression analysis. To identify the potential transcription factors (TFs) that control expression of the gene regulatory networks they have done SCENIC analysis which is a popular method to do so.

Initial figures present the transcriptional data derived from analysis of fetal CD140a+ and A2B5+ cells (fig 1 and 2) and then of adult A2B5+ cells (Fig 3). These indicate the lineage related differences of these cell types (module Fig 3) reflecting the concerns raised above regarding comparing them with regard to age related senescence changes.

Data in Fig 4 describes the transcription factors (TFs) linked to regulating the observed differences in cell cycling and senescence and the signaling pathways being regulated.

Fig 5 presents the studies validating the prediction that the adult repressors identified (E2F6, ZNF274, MAX, IKZF3) could induce age associated gene expression changes in young hGPCs derived from an ipsc line (selected based on its transcriptome being close to that of primary hGPCs). The authors use a lenti-virus based doxycycline inducible overexpression system for each of the TFs. The obtained molecular patterns do model what was seen in the comparison of the early and adult primary GRPs.

What was the state of differentiation of the ipsc cells used – were they all expressing CD140a under base-line conditions, matching the properties of the primary early GRPS. Would more differentiated cells be available that could model the adult A2B5+ cells?

PDGFRa was amongst genes down-regulated following viral induction; did the overall pattern of changes in gene expression suggest that the cells were differentiating along the OL lineage?

Fig 6 presents the studies aimed at identifying the post transcriptional regulators of gene expression in early and adult GRPs. Affymetrix Gene Chip arrays were used to obtain miRNA expression profiles. The data analysis supports the hypothesis that miRNAs mediated regulation complements the direct transcription factor regulation. Future work can determine the effect of manipulating specific miRNAs (would parallel the functional work done with the direct TFs).

The final section in the results indicates the availability of an interactive data base that allows independent exploration of the data presented in this communication. This reviewer views this as a very positive contribution as one the current report has such extensive data that would preclude commenting on each gene, TF, and miRNA mentioned by the authors.

Overall, this report contains an extensive analysis of how one can identify potential direct and indirect regulators of cycling and senescence in OL lineage cells and confirm the functional effects of the molecular predictions. The central concern with regard to the application of the data to the question of age related deficits in human remyelination is that the comparative studies are conducted in non-identical cell types and do not cover the human aging spectrum (the study covers fetal vs young adult).

Reviewer #3 (Remarks to the Author):

The manuscript “Repression of developmental transcription factor networks triggers aging-associated gene expression in human glial progenitor cells” investigates transcriptomic networks regulated during aging in human glial progenitor cells. To assess these pathways, extensive RNA-seq and sc-RNAseq experiments coupled with lentiviral overexpression of identified TFs was performed. Differentially expressed genes between varying conditions were calculated and annotated to assess their functional properties. The manuscript specifically highlights TF actions related in the regulation of proliferation capacity between fetal and adult subjects. Overall, the manuscript is well-written, and all experiments are explained and structured properly. I have a few concerns related to negative control used and slight over-interpretation regarding the senescent state, which prevent me from suggesting the manuscript for publication in Nature Communications at this stage. I also feel that the scRNAseq data was not exploited enough to test some of their TF/miRNA network hypotheses. Below please find my specific points to help improve the manuscript before publication.

Major Points

1. The authors use un-transduced, doxycycline-treated cells as negative control in the experiments of Figure 5. This seems sub-optimal, because one would expect that transducing cells with lentiviral particles offers a greater source of variance (cellular stress) than the doxycycline treatment. It seems plausible that lentiviral transduction alone could lead to the decrease in pathways affecting proliferation, and I feel that transduced cells, but not treated with dox, would have been the more appropriate negative control. Without further explanation, it remains unclear why the authors did not use the EGFP-only vector that they used later in the sc-RNA experiment also in the bulk sequencing, and I suggest to include experimental data clarifying to what extent cells transduced with an EGFP-only virus show a reduction in cell proliferation pathways.
2. The authors deeply discuss cellular senescence in the context of decreased cell proliferation, which is indeed one major hallmark of senescence. However, I feel that the data does not suffice to state that these cells are in a senescence state without further characterizing the phenotype of senescence with other functional experiments. Here they could analyze cells that did not receive any virus in the overexpression experiment and investigate, how these cells affect their surrounding in the dish (SASP) or perform well-established assays on markers like b-Gal or p16 expression in these cells in cultures. Only on transcriptomic signatures, I would not necessarily believe, that these cells really are in a senescent state.
3. In Fig. 4, the authors present a transcription factor activity prediction of fetal or adult GPCs, but they don't follow up with a testing of their network. This experiment could be improved to validate the prediction, and to bolster the conclusions, using the scRNAseq data. One strategy could be to select the cells that express highest levels of the identified TFs and see if the downstream targets are upregulated on a cell-by-cell basis.

4. Related to my previous point, the authors should infer the activity of their miRNA targets from the scRNAseq. Downregulation of genes that have a miRNA target sequence from their small RNA screen would be a convincing independent dataset to validate which miRNAs might be most active in GPCs.

5. In Figure 2H, the UMAP clearly shows heterogeneity in the GPC population. These differences between populations of cells might be of great value for the authors points, if some have high MYC TF expression or senescence gene expression for example. The authors should isolate and then re-cluster this population separately and look for evidence for their identified TF and miRNA groups.

6. For FACS-sorted whole brain analysis, capturing 1,000 cells seems very little, and maybe not enough to draw very strong conclusions. With no directly comparable data at hand with more cells (e.g. 5,000-10,000 cells), could the authors at least elaborate the choice of cell numbers, and evaluate the power of their findings to other comparable studies. I feel that this would be helpful for following studies, and elevate transparency

Minor Points

Could the authors explain, why the variance (PC1) in CD140a+ cells is even slightly higher than when comparing these cells to the other marker (Fig.1B). The authors address by only comparing common DEGs between both markers, but still, the variance is quite high. Could the authors please include an analysis/plot showing the individual donors on this PCA?

According to the sc-UMAP in Fig.2, it looks like that only a small fraction of the captured cells are GPCs. Could the authors add a plot that quantifies the cell type relative proportions. I feel this is important to get a feeling on how well intersecting genes in the bulk RNA-sequencing resemble pathways affected in GPCs, or if they are mainly conserved pathways in NPCs, since these look like make up the biggest proportions of these cells.

Typo in line 271 ☒ doxycycline

RSEM version is missing in line 496

The < and > symbols on line 118 should be flipped?

Can the authors please comment on the adult loss of CD140 expression?

The authors first ran DESeq2 and then filtered for expressed genes, and I am worried that this may affect vst-normalizing. DESeq2's standard workflow includes filtering based on rawcounts, but they didn't mention it here. I don't think it will affect the outcome too much, but it should be explained.

Comments about the potential therapeutic value of the findings should be toned-down, as they are based on iPSC based lentiviral overexpression in one specific cell type in vitro. The data is interesting, but too basic in my opinion to raise translational value.

How did the authors extract 'single cell co-expression data' (line 143)? There is no explanation, and in the methods, they only state they use genes present in 1% of the cells. Please elaborate

To the referees:

Thank you for the reviews of our paper, “Repression of developmental transcription factor networks triggers age-associated senescence in human glial progenitor cells.” We have revised and supplemented the paper substantially, so as to address all concerns and requests. In particular, we have consolidated the previous Figures 1 and 2 into a **new Figure 1**; and we have added substantial new data to **Figures 4 and 5, as well as to Extended Data Figures 1, 4, 5, and 6, and Extended Data Tables 4 and 5.**

We would like to respond to the specific comments and questions as follows:

Reviewer #1 (Remarks to the Author):

In this manuscript the authors use RNA sequencing to characterize the changes that occur in oligodendrocyte/glia precursor cells with age. Through a combination of cell specific bulk and single cell sequencing they show that adult human glial precursors have significantly distinct profiles from their fetal counterparts. Analysis of the differences between fetal and adult purified cell population revealed a surprisingly larger number of transcriptional differences. Pathway and network analyses identify regulators of MYC signaling as a central element and overexpression of transcriptional repressors alters relevant gene expression in iPSC derived cell lines. A number of miRNAs are also identified that may contribute to regulation of a mature glial phenotype.

This is an interesting paper that provide new insights into the maturation of oligodendrocytes in the human CNS. Previous studies have suggested that signaling through the MYC pathway is an important regulator of OPC aging and the current manuscript contains significant new data that extends and support this hypothesis. While the central take-home message of the manuscript is relatively straight forward, the current paper includes a significant amount of data and studies that are not directly relevant to control of OPC/GPC gaining. This includes much of the data in Figure 1 and 2. Indeed the A2B5/CD140a comparators are somewhat confusing and not further explored in the manuscript. A somewhat shortened and more focused manuscript would be easier to follow and potentially more impactful.

Figures 1 and 2 have now been simplified and merged into a single figure to be more concise. The text has been streamlined as well for clarity.

One general concern with the current paper is that while there is a lot of transcriptional analyses that implicate both a repressor network and the role of a number of miRNAs in the aging of glial precursors, there is very limited experimental data to demonstrate the functionality of the proposed pathways. Such validation may be beyond the scope of the current paper, but the authors should temper their statements in its absence.

We have added two datasets that greatly increase our validation of the repressor network described in the current study. First, we determined functional output of the senescence pathways, and found a significant increase in p16/CDKN2A immunolabelling by GPCs transduced to express any one of our 4 principal candidate genes, E2F6, IKZF3, MAX, or ZNF274. Concordantly, we also noted a significant increase in p21/CDKN1A expression by cells overexpressing E2F6 or ZNF274. We have added these data as a new **Figure 4E**. In addition, we conducted qPCR against the most prominent downstream targets of E2F6 and ZNF274 identified in our single-cell RNA-seq datasets, and found that overexpression of each of these repressors indeed resulted in the significant down-regulation of its targets. These data have also been added, as a new **Figure 5I**.

Some other minor issues:

1. Related to the comment above, in Figure 5 the authors show the effects of expressing transcriptional repressors in iPSC derived hGPCs. The data suggests the induction of an aged transcriptome but make no comment on the proliferative or morphological characteristics of the cells. Specifically, it would be interesting to know if there was a reduction in cell division and an increased tendency to differentiate in the treated cells.

The transcriptional data from these experiments were consistent with a sharp reduction in cell division, via the downregulation of active proliferation transcripts. MKI67 in particular was significantly suppressed in the E2F6-transduced cells. We followed up on this transcriptional observation by staining for KI67 the mitotic marker in doxycycline-inducible E2F6 and ZNF274 overexpressing GPCs, and found them to have significantly lower expression – i.e., lower mitotic indices – than cells infected with either a doxycycline-inducible EGFP control virus or treated with doxycycline alone. These data have been added to the text, as well as to a new **Figure 5K**. As noted, we also observed the acquisition of markers of senescence in these cells, including not only p21 and p16, but the significant increase in beta-galactosidase activity as well, in E2F6-overexpressing cells (**Fig. 5L**).

2. The authors should attempt to reduce the degree of abbreviations throughout. Examples include defining the different cell types discussed on page 4. What is a pre-GPC? It is never articulated.

Done. We have spelled out less-used abbreviations whenever possible, and across the board have better described cell definitions throughout the manuscript. Pre-GPCs comprise a transitional stage harboring transcripts of both uncommitted neural progenitors and glial progenitors, and cluster distinctly from those more sharply-delineated phenotypes.

Overall, this is an interesting and well-done paper that will be of significant interest to workers in the field. Some refinement of the current manuscript would likely enhance its impact.

We thank the reviewer for his/her kind remarks and helpful suggestions.

Reviewer #2 (Remarks to the Author):

Mariani et al provide a study aimed at understanding why there is reduced re-myelination with aging taking advantage of their access to primary human OL lineage cells and OL lineage cells derived from ipscs. The main message is derived by comparisons of the transcriptomes of CD140a+ (PDGFR+) cells derived from 17-20 week human fetal material and A2B5+ cells derived from adult human brain (ages 18-27). They identify age related transcriptional changes that repress genes involved in mitotic expansion and provide validation of the contribution of these genes by assessing effects of their overexpression in GPCs derived from an ipsc line. Establishing that these regulatory pathways contribute to age related myelination failure would be a significant contribution.

We thank the referee for noting the importance of this work.

Overall challenges to answering why there is inadequate remyelination in acquired human disease include what is the OL lineage cell most responsible for such a process and how is it impacted by aging. The comparative transcriptome focused studies involve CD140a+ (PDGFR+) cells derived from 17-20 week human fetal material and A2B5+ cells derived from adult human brain (ages 18-27). Thus these comparisons involve cells at different stages of the OL lineage rather than the same cell young versus older adults. The materials used in the current study do not allow for more matched comparisons.

The antigenic phenotype of the human GPC, and its relative specificity with regards to other brain phenotypes, is a moving target as development proceeds – reflecting the maturation of both this cell type, and of its neighbors. Basically, the fetal GPC is not the same as its adult descendent and functional

counterpart; fetal and adult human GPCs do not express the same antigenic repertoires as they mature, nor do those markers have the same cell type-specificity at different ages. Whereas A2B5, defining a set of GPC-specific gangliosides, appears expressed only by GPCs in the adult brain, A2B5 is also expressed by young neurons in fetal brain. Hence our concurrent depletion of PSA-NCAM defined cells, which include young neurons. However, PSA-NCAM can be expressed by young GPCs as well, resulting in a drop in GPC yield and selection bias upon its immunodepletion. A more specific marker of GPC phenotype in development is instead the PDGF α receptor and its CD140a epitope - which is also expressed by most A2B5⁺/PSA-NCAM⁻ cells. However, PDGF α R expression is linked to both activity and mitotic expansion, so that its expression falls with maturation; we have found that most adult human GPCs do not express sufficiently high surface levels of PDGF α R/CD140a to be sorted on its basis. Given then the lack of specificity of A2B5 expression by fetal cells, and the relative loss of PDGF α R expression by their adult-derived counterparts, we found that the optimal comparison by which to define the developmental course of the human GPC was the intersection of CD140a⁺ and A2B5⁺/PSA-NCAM⁻ (fetal) vs A2B5⁺ (adult). Our initial **Figure 1** laid out the logic and data behind this argument, but in what we now realize was a confusing manner. **We have thus now consolidated Figures 1 and 2 into a single new, and clearer, Figure 1**, which notes the basis for this comparison while better describing our experimental design and analytic pipeline.

The authors show that CD140a⁺ is down regulated on the adult A2B5⁺ progenitor cells isolated from the human adult brain specimens and that no CD140a⁺ population is derived from this material. In situ analyses of human brain including by the current authors (Osorio GLIA 2023) and the Allen brain atlas indicate that CD140a⁺/NG2 cells are abundant in adult human brain and thus the issue whether current techniques are inadequate to recover such cells. Perhaps tissue based analytic techniques will be needed. The converse of isolating equivalent A2B5⁺ cells is also problematic as A2B5⁺ cells in the fetus are not yet all OL lineage committed; the adult A2B5⁺ cells express later lineage markers including MBP and are not present in the fetal material. If the primary aim of the study is to assess proliferation and senescence related mechanisms as a function of aging, it would seem more informative if one compared younger and older adults as done in rat related studies such as done by Neumann et al using A2B5⁺ cells. Late lineage cells are rarely present in fetal material (equivalent of adult A2B5⁺ cells).

While the expression of PDGFRA mRNA by adult GPCs is detectable, as noted above, it is both activity and ligand-dependent; the level of PDGF α R/CD140a protein expression by adult human GPCs is minimal. We've looked extensively in human tissue, chimeric brain, and across a variety of methodologies, and this has been a consistent observation. The Osorio paper referred to by the referee probed NG2, not PDGF α R, and NG2 has proven difficult to sort on (CSPG4 is highly sensitive to essentially all dissociation conditions we've tried, and it's not specific to hGPCs in any case, as it is highly expressed by pericytes); also, the Allen Brain Atlas data referred to by the referee included in situ, not immunolabels. The Neumann and Franklin work was done in rats, was really directed at a different point, and for obvious reasons of species choice did not span an age range as wide as ours (fetal vs mid-20s); our data sets are certainly complementary, but there is no better choice for studying human glial biology than to use human cells.

The bulk and scRNAseq data of the manuscript are analyzed using standard tools and approaches. Specifically, the authors have taken into account the effect of different batches and included this in their differential gene expression analysis. To identify the potential transcription factors (TFs) that control expression of the gene regulatory networks they have done SCENIC analysis which is a popular method to do so.

Initial figures present the transcriptional data derived from analysis of fetal CD140a⁺ and A2B5⁺ cells (fig 1 and 2) and then of adult A2B5⁺ cells (Fig 3). These indicate the lineage related differences of these cell types (module Fig 3) reflecting the concerns raised above regarding comparing them with regard to age related senescence changes.

Data in Fig 4 describes the transcription factors (TFs) linked to regulating the observed differences in cell cycling and senescence and the signaling pathways being regulated.

Fig 5 presents the studies validating the prediction that the adult repressors identified (E2F6, ZNF274, MAX, IKZF3) could induce age associated gene expression changes in young hGPCs derived from an ipsc line (selected based on its transcriptome being close to that of primary hGPCs). The authors use a lenti-virus based doxycycline inducible overexpression system for each of the TFs. The obtained molecular patterns do model what was seen in the comparison of the early and adult primary GRPs.

What was the state of differentiation of the ipsc cells used – were they all expressing CD140a under base-line conditions, matching the properties of the primary early GRPS. Would more differentiated cells be available that could model the adult A2B5+ cells?

Roughly 85% of iPSC GPCs (781/921) had detectable PDGFRA expression via scRNA-seq; after accounting for false negative non-detection of PDGFRA transcripts using 10x Genomics v3, we think that the iPSC GPCs we used faithfully recapitulated the phenotype of bona fide young human GPCs. These cells also express A2B5 (as we have previously reported, in Sim et al., Nature Biotech., 2011), but that marker does not provide the same high degree of specificity in these iPSC cultures, any more than it does in fetal human brain; both contain residual neural progenitors and fibrous astrocytes that also express A2B5, unlike adult brain tissue, in which A2B5's expression is more clearly restricted to GPCs.

PDGFRA was amongst genes down-regulated following viral induction; did the overall pattern of changes in gene expression suggest that the cells were differentiating along the OL lineage?

In the two overexpression conditions we evaluated, including ZNF274 and E2F6 transduction, we did not note any transcriptional changes that suggested differentiation. Rather, transcripts associated with cyclicity were sharply suppressed, and markers associated with a loss of mitotic competence (e.g., p21) were increased, suggesting some de-linking of mitotic competence and activity from phenotypic differentiation.

Fig 6 presents the studies aimed at identifying the post transcriptional regulators of gene expression in early and adult GRPs. Affymetrix Gene Chip arrays were used to obtain miRNA expression profiles. The data analysis supports the hypothesis that miRNAs mediated regulation complements the direct transcription factor regulation. Future work can determine the effect of manipulating specific miRNAs (would parallel the functional work done with the direct TFs).

The final section in the results indicates the availability of an interactive data base that allows independent exploration of the data presented in this communication. This reviewer views this as a very positive contribution as one the current report has such extensive data that would preclude commenting on each gene, TF, and miRNA mentioned by the authors.

We thank the referee for recognizing the potential import of this database, which we believe will prove a significant and unique contribution of this paper. Since our initial submission of this manuscript, we have further updated Glial Explorer to load and survey single cell data even more rapidly. We are confident that this will prove an important resource to the field.

Overall, this report contains an extensive analysis of how one can identify potential direct and indirect regulators of cycling and senescence in OL lineage cells and confirm the functional effects of the molecular predictions. The central concern with regard to the application of the data to the question of age-related deficits in human remyelination is that the comparative studies are conducted in non-identical cell types and do not cover the human aging spectrum (the study covers fetal vs young adult).

We thank the referee for his/her appreciation of the significance of this work. We trust that the referee will appreciate our point that the fetal and adult hGPCs, while lineally-related and of comparable function and differentiation potential, comprise different and dynamic, stage-specific and temporally-derived phenotypes; there's really no such thing as a truly identical glial progenitor cell phenotype in the fetal and adult human brain (indeed, it's the point of this study to define the differences in hGPC gene expression between the two, hence no reason to have assumed that their surface antigenicities would be completely identical).

Reviewer #3 (Remarks to the Author):

The manuscript "Repression of developmental transcription factor networks triggers aging-associated gene expression in human glial progenitor cells" investigates transcriptomic networks regulated during aging in human glial progenitor cells. To assess these pathways, extensive RNA-seq and sc-RNAseq experiments coupled with lentiviral overexpression of identified TFs was performed. Differentially expressed genes between varying conditions were calculated and annotated to assess their functional properties. The manuscript specifically highlights TF actions related in the regulation of proliferation capacity between fetal and adult subjects. Overall, the manuscript is well-written, and all experiments are explained and structured properly. I have a few concerns related to negative control used and slight over-interpretation regarding the senescent state, which prevent me from suggesting the manuscript for publication in Nature Communications at this stage. I also feel that the scRNAseq data was not exploited enough to test some of their TF/miRNA network hypotheses. Below please find my specific points to help improve the manuscript before publication.

Major Points

1. The authors use un-transduced, doxycycline-treated cells as negative control in the experiments of Figure 5. This seems sub-optimal, because one would expect that transducing cells with lentiviral particles offers a greater source of variance (cellular stress) than the doxycycline treatment. It seems plausible that lentiviral transduction alone could lead to the decrease in pathways affecting proliferation, and I feel that transduced cells, but not treated with dox, would have been the more appropriate negative control. Without further explanation, it remains unclear why the authors did not use the EGFP-only vector that they used later in the sc-RNA experiment also in the bulk sequencing, and I suggest to include experimental data clarifying to what extent cells transduced with an EGFP-only virus show a reduction in cell proliferation pathways.

To address the referee's concern, we directly assessed the effect of lentiviral transduction on hGPC proliferation by staining for Ki67 in both of the controls used in this study, hGPC cultures treated with doxycycline alone, or those first infected with an inducible EGFP virus followed by doxycycline. In all cases, hGPCs in which E2F6 or ZNF274 were lentivirally overexpressed had significantly lower mitotic indices, as scored by Ki67 expression, than did their respective controls. (Interestingly, at 7 days following treatment, we noted a small increase in proliferation in the EGFP controls relative to doxycycline treatment alone.) We have added these data as a new **Fig. 5K**. Furthermore, we also assayed both controls for beta-galactosidase activity, and found no increase in either control, whereas E2F6 overexpression increased the beta-gal incidence relative to both controls. We have added these new data as a new **Fig. 5L**, and have added the following text to the Results as well (page 10):

We next assessed the mitotic index of these cells via immunolocalization of Ki67, the protein product of MKI67, in transcription factor-overexpressing hGPCs, vs both Dox-alone controls, and EGFP expressing CTR hGPCs (Fig. 5K). We found that both E2F6 and ZNF274 overexpression significantly reduced Ki67 incidence when compared to either the dox alone or EGFP control. Further, we assessed these cultures for the induction of beta-galactosidase activity, which is typically observed in aged cells. We found that hGPCs transduced to express E2F6 exhibited a significant upregulation of beta-galactosidase activity relative to both the EGFP-only virus and

Dox-alone controls (Fig. 5L). Together, these data revealed downstream transcriptional and phenotypic changes in response to E2F6 or ZNF274 overexpression, which recapitulated salient features of aging by human GPCs, and which thereby supported our predicted network for the regulation of aging-associated transcription by human GPCs.

2. The authors deeply discuss cellular senescence in the context of decreased cell proliferation, which is indeed one major hallmark of senescence. However, I feel that the data does not suffice to state that these cells are in a senescence state without further characterizing the phenotype of senescence with other functional experiments. Here they could analyze cells that did not receive any virus in the overexpression experiment and investigate, how these cells affect their surrounding in the dish (SASP) or perform well-established assays on markers like b-Gal or p16 expression in these cells in cultures. Only on transcriptomic signatures, I would not necessarily believe, that these cells really are in a senescent state.

We thank the referee for this suggestion. We went back and analyzed the *uninfected* GFP-negative cells that were in the same cultures as successful GFP⁺ transductants. In particular we stained for p16 and p21 at one week after infecting hGPC cultures with lentiviruses expressing either E2F6, MAX, IKZF3, or ZNF274; we then assessed the proportions of p16⁺ and p21⁺ cells among the GFP⁺ and GFP⁻ cells of each infected culture. We found that all four repressors induced significant increases in p16 in the GFP⁺ cells, relative to the GFP-only controls. In addition E2F6 and ZNF274 also significantly increased the proportion of p21⁺ cells among the GFP⁺/p21⁺ cells (**Fig. 4E**). Importantly though, and directly to the referee's question, we found that p16 (though less so p21) was indeed also significantly increased in uninfected, GFP negative cells in those cultures overexpressing E2F6, MAX, and ZNF274; this suggested a cell-extrinsic effect, consistent with the acquisition of a SASP phenotype by the transduced cells. These data have been added to both the text and to a new **Extended Data Fig. 4B**.

To confirm the acquisition of an aged phenotype, we also assessed both the Ki67-scored mitotic index and β -galactosidase expression by E2F6 and ZNF274 transduced hGPCs, at 7 days after infection. We found that both transcription factors significantly suppressed the mitotic index, and that E2F6 overexpression in particular significantly increased the proportion of β -gal⁺ cells, again consistent with the potentiation of an aged phenotype. As noted above in the context of discussing our controls, we have added **new Figures 5K and 5L** with these latter data, and new text to the Results (page 10, as noted above in item 1).

3. In Fig. 4, the authors present a transcription factor activity prediction of fetal or adult GPCs, but they don't follow up with a testing of their network. This experiment could be improved to validate the prediction, and to bolster the conclusions, using the scRNAseq data. One strategy could be to select the cells that express highest levels of the identified TFs and see if the downstream targets are upregulated on a cell-by-cell basis.

We have now done so. To validate the networks identified from our *in vivo* bulk RNA-sequencing, we applied our approach for identifying transcriptional networks based on differential gene expression to our scRNA-Seq data, following overexpression of our candidate TFs. We have included these data in new **Figures 5C, D, E and H**, and a new table in **Extended Data Table 4**, and have added the following paragraph of text to the Results (page 9):

We next mined these genesets to validate the E2F6 and ZNF274 adult repressor targets identified from our *in vivo* bulk RNA-sequencing dataset (Fig. 3G, Extended Data Table 3). rCisTarget²⁶ predicted numerous enriched motifs attributed to E2F6 or ZNF274 repression in their respective overexpression genesets with largely significant enrichment for predicted targets in repressed rather than activated genes (E2F6: $p < 2.2 \times 10^{-16}$, ZNF274: $p = 4.5 \times 10^{-11}$, Fisher's exact test, Extended Data Table 4). Of these predicted repressed targets that were also found to be repressed in adult GPCs, all of the E2F6 targets from our *in vivo* bulk analysis were recovered via our scRNA-seq analysis, along with an additional 33 repressed direct targets (Fig. 5C). In the case

of ZNF274, our scRNA-seq analysis recovered all but 5 predicted direct targets from our in vivo bulk analysis, and added another 6 prospective direct targets (Fig. 5D). Next, we used AUCell²⁶ to score the activity of both regulons uncovered in our scRNA-Seq datasets of each group. This revealed the significantly repressed signatures of the E2F6 and ZNF274 regulons in their respective overexpression paradigms vs EGFP controls, as well as in each other's paradigm, due to their mutual repression of shared targets (E2F6: $p < 2.2 \times 10^{-16}$; ZNF274: $p < 2.2 \times 10^{-16}$; Wilcoxon rank sum test) (Fig. 5E).

In addition, we sought to validate six of these predicted direct target genes via qPCR (ETV1, HIST1H1D, HMGB3, IDH2, NUSAP1, PEG10); we have included these data in a new **Figure 5I**, with this additional text:

Among these curated genes, many were identified as direct targets of E2F6 and/or ZNF274 in addition to being similarly differentially expressed in primary GPCs during aging (Fig. 5H). Of those targeted by E2F6 and/or ZNF274 that were also differentially expressed in primary GPCs during aging, we validated via qPCR the repression of ETV1, NUSAP1, and PEG10 in both conditions and HMGB3 following E2F6 repression vs. an EGFP control virus (Fig. 5I).

4. Related to my previous point, the authors should infer the activity of their miRNA targets from the scRNAseq. Downregulation of genes that have a miRNA target sequence from their small RNA screen would be a convincing independent dataset to validate which miRNAs might be most active in GPCs.

We have now done so, by analyzing the differentially expressed genesets from our E2F6 and ZNF274 overexpression studies for the enrichment of fetal- or adult-GPC enriched miRNAs and their targets (**Fig. 5**). Among those genes downregulated in response to E2F6 or ZNF274 overexpression, none were predicted targets of adult-enriched miRNAs (**Extended Data Table 5**). In contrast, among those genes upregulated in either E2F6- or ZNF274-transduced GPCs, a number were found to be targeted by fetal GPC miRNAs (**Extended Data Fig. 5C**). These data suggest that the expression of aging-associated, adult GPC gene sets may, at least in part, be facilitated via the E2F6 or ZNF274 mediated repression of fetal miRNAs that might otherwise suppress the expression of these aging-associated transcripts. Of particular note in this regard, E2F6 was predicted to repress a group of these fetal-enriched GPC miRNAs in our transmiR analysis (**Extended Data Fig 5B**); these included miR-106a-5p, miR-130a-3p, miR-130b-3p, miR-92b-3p, miR17-5p, miR-363-3p, and miR-504-5p.

While we believe that further functional exploration of these miRNA networks is beyond the scope of this paper, we thank the referee for raising this question, which will surely spark later investigation.

5. In Figure 2H, the UMAP clearly shows heterogeneity in the GPC population. These differences between populations of cells might be of great value for the authors points, if some have high MYC TF expression or senescence gene expression for example. The authors should isolate and then re-cluster this population separately and look for evidence for their identified TF and miRNA groups.

Any heterogeneity in the hGPC population would take a much larger population to meaningfully assess; there's actually little heterogeneity in this population, with relatively minimal variability in differentiated state. More broadly, our sense is that whatever heterogeneity exists in the population shown in (the prior version of) **Fig. 2H** reflects stage-defined states rather than reflecting fundamentally cell-intrinsic phenotypic diversification, and hence comprise transient events. While the referee's suggestion is an interesting one, it would take a much larger study, that would seem unlikely to change any of the fundamental conclusions of this study, while being beyond our intended scope.

6. For FACS-sorted whole brain analysis, capturing 1,000 cells seems very little, and maybe not enough to draw very strong conclusions. With no directly comparable data at hand with more cells (e.g. 5,000-10,000 cells), could the authors at least elaborate the choice of cell numbers, and evaluate the power of

their findings to other comparable studies. I feel that this would be helpful for following studies, and elevate transparency.

We aimed to capture 1,000 cells for both A2B5⁺/PSA-NCAM⁻ and CD140a⁺ sorts as a means of validating the composition of these sort modalities, which we then interrogated more deeply via bulk RNA-seq in **Figure 1**. This number was chosen simply to distinguish cell populations with a minimum of 20 cells at a 1% occurrence. This yielded 1,053 A2B5⁺/PSA-NCAM⁻ and 957 CD140a⁺ high quality cells, whose relative abundances are now provided in **Extended Data Figure 1E**.

Minor Points

Could the authors explain, why the variance (PC1) in CD140a⁺ cells is even slightly higher than when comparing these cells to the other marker (Fig.1B). The authors address by only comparing common DEGs between both markers, but still, the variance is quite high. Could the authors please include an analysis/plot showing the individual donors on this PCA?

A labeled PCA has been added in **Figure 1B**.

According to the sc-UMAP in Fig.2, it looks like that only a small fraction of the captured cells are GPCs. Could the authors add a plot that quantifies the cell type relative proportions. I feel this is important to get a feeling on how well intersecting genes in the bulk RNA-sequencing resemble pathways affected in GPCs, or if they are mainly conserved pathways in NPCs, since these look like make up the biggest proportions of these cells.

This plot has been added in **Extended Data Figure 1D**.

Typo in line 271 ◊ doxycycline

Fixed, thanks.

RSEM version is missing in line 496

Added.

The < and > symbols on line 118 should be flipped?

We have improved clarity to explain the criteria for low-quality cell filtering.

Can the authors please comment on the adult loss of CD140 expression?

The drop in OPC expression of PDGFRA with aging has been noted previously, both by our group and others, including in human (PMID 34952986), mouse (PMID 36044851), and rat (PMID 32434922). We have added this text, with a number of additional relevant citations, to the Results (page 5):

We have previously noted that A2B5 selection is sufficient to isolate GPCs from adult human brain¹⁸, and is more sensitive in adults than selection based on CD140a/PDGFaR, since PDGFRA expression falls with maturation and is typically lowly expressed by adult hGPCs^{12,14,18,20,27}.

The authors first ran DESeq2 and then filtered for expressed genes, and I am worried that this may affect vst-normalizing. DESeq2's standard workflow includes filtering based on rawcounts, but they didn't mention it here. I don't think it will affect the outcome too much, but it should be explained.

Filtration of genes based on abundance was only done for downstream analyses here. VST-normalization and differential expression were carried out on raw counts of all genes without pre-filtering. Pre-filtering is not necessary according to the authors of DESeq2 (<http://bioconductor.org/packages/devel/bioc/vignettes/DESeq2/inst/doc/DESeq2.html#pre-filtering>). This has been clarified in the text (page 16):

DE analysis was carried out in DESeq2¹⁵⁷ on all genes where paired analyses (Fetal A2B5⁺ vs CD140a⁺, fetal CD140a⁺ vs CD140a⁻) had paired information added to their models.

These data were then further filtered by meaningful abundance, defined as a median TPM (calculated via RSEM) of 1 in at least 1 group (20,663 genes met this criterion prior to DE) for downstream analyses (IPA, Networks).

Comments about the potential therapeutic value of the findings should be toned-down, as they are based on iPSC based lentiviral overexpression in one specific cell type in vitro. The data is interesting, but too basic in my opinion to raise translational value.

Fair enough. We've toned down relevant statements accordingly.

How did the authors extract 'single cell co-expression data' (line 143)? There is no explanation, and in the methods, they only state they use genes present in 1% of the cells. Please elaborate.

We established single cell co-expression in the initial steps of the standard SCENIC pipeline, by first filtering out lowly detected genes (Expressed in <1% of cells) and then analyzing co-expression and co-regulatory activity of single cell data in the subsequent GENIE3 analytic step, which infers gene-regulatory networks from single cell expression data. To clarify this in the text, we have added the following to the Methods section (page 17):

Prediction of active transcription factor regulons was carried out with SCENIC in R²⁵, using the standard workflow and hg38 databases located at <https://resources.aertslab.org/cistarget/>. Genes were included in co-expression analyses if they were expressed in at least 1% of cells before the GENIE3 step.

Thank you all for your helpful suggestions and efforts on behalf of this paper, which we believe have strengthened it significantly.

REVIEWERS' COMMENTS

Reviewer #1 (Remarks to the Author):

In this revised manuscript the authors have responded very effectively to the concerns raised in the original reviews. The revised manuscript contains a significant amount of new data that provides strong support for the original concept that suppression of developmental transcription pathway contribute to generation of mature OPCs. The work is well done and comprehensive. Given that this is a relatively unexplored area this manuscript will provide a solid base for further investigation. I have no additional concerns.

Reviewer #2 (Remarks to the Author):

For this reviewer, a clear message of the report comes from the detailed functional molecular studies carried out on the ipsc derived cells that have the properties of early progenitors and that define regulatory mechanisms impacting on senescence. The authors were a key group in defining the properties of early progenitors (PDGFR /NG2) in fetal human brain and A2B5 cells from adults. The current paper uses a comparison of their molecular profiles to identify differential expression of senescent genes in these 2 cell types leading to the very worthwhile studies to identify pathways and transcription factors and miRNAs regulating senescence.

However, still less clear from the manuscript and rebuttal letter is how the findings relate to what occurs with cells at the same OL lineage stage during the human aging process – all are present in the human brain. Comparative functional and molecular studies within cell types may not have been possible as one cannot derive adult type A2B5 +cells from fetal tissue and cannot derive pDGFR+ cells from adult samples. Perhaps this is not the central aim of the report nor is an overall comparison between different subtypes (reviewer 1 - Indeed the A2B5/CD140a comparators are somewhat confusing and not further explored in the manuscript).

Some of the terminology used is not entirely clear.

As stated in the introduction – “Glial progenitor cells (GPCs, also referred to as oligodendrocyte progenitor cells or NG2 cells) emerge during the 2nd trimester to colonize the human brain, and persist in abundance throughout adulthood”. The A2B5+ cells isolated from adults have little PDGFR (“PDGFRA in adult A2B5+ hGPCs was expressed with a median TPM of only 0.55, compared to its median TPM of 47.56 in fetal A2B5+ cells). One asks whether these adult A2B5+ cells referred to by some as pre-oligodendrocytes are not the same population as GRCs. As indicated GPCs are present in the adult human brain but have not been successfully isolated as viable cells for functional studies.

The majority of ipsc cells studied, as stated in the rebuttal letter, were PGDFR+ indicating they were early in the lineage – the study did not intend to address a comparison of early and late progenitors to match the analysis of primary fetal PDGFR and adult A2B5 cells. The ipsc studies do not address properties of cells from old and young donors.

Perhaps most of the above concerns can be addressed in the discussion with a focus on the fundamental regulatory pathways identified and leave open the question how they impact on each cell type during human aging.

Author's Response to Reviewers

Reviewer #1 (Remarks to the Author):

In this revised manuscript the authors have responded very effectively to the concerns raised in the original reviews. The revised manuscript contains a significant amount of new data that provides strong support for the original concept that suppression of developmental transcription pathway contribute to generation of mature OPCs. The work is well done and comprehensive. Given that this is a relatively unexplored area this manuscript will provide a solid base for further investigation. I have no additional concerns.

We thank the referee for his/her comments, and for recognizing the importance of the central thesis of the paper.

Reviewer #2 (Remarks to the Author):

For this reviewer, a clear message of the report comes from the detailed functional molecular studies carried out on the ipsc derived cells that have the properties of early progenitors and that define regulatory mechanisms impacting on senescence. The authors were a key group in defining the properties of early progenitors (PDGFR /NG2) in fetal human brain and A2B5 cells from adults. The current paper uses a comparison of their molecular profiles to identify differential expression of senescent genes in these 2 cell types leading to the very worthwhile studies to identify pathways and transcription factors and miRNAs regulating senescence.

However, still less clear from the manuscript and rebuttal letter is how the findings relate to what occurs with cells at the same OL lineage stage during the human aging process – all are present in the human brain. Comparative functional and molecular studies within cell types may not have been possible as one cannot derive adult type A2B5 +cells from fetal tissue and cannot derive pDGFR+ cells from adult samples. Perhaps this is not the central aim of the report nor is an overall comparison between different subtypes (reviewer 1 - Indeed the A2B5/CD140a comparators are somewhat confusing and not further explored in the manuscript).

We thank the referee or this comment, which is correct; we did not attempt in this paper to parse the subtypes of glial progenitors present in the human brain, or to follow any give subtype over time. We believe this would have been a fundamentally futile effort, since the transcriptional state of any given oligodendroglial lineage is a moving target as maturation and aging proceed, such that the canonical markers of a phenotype at one point in differentiation may not be so at an earlier of later stage. This is indeed exemplified by the differential expression of a marker – PDGF α R – by otherwise functionally analogous GPCs at fetal and adult stages. To amplify this point further, we have added the following to the **first paragraph of the Discussion**:

Bipotential oligodendrocyte-astrocyte glial progenitor cells were best identified by PDGF α R/ CD140a expression in fetal development; these cells were largely captured by the A2B5⁺/PSA-NCAM⁻ phenotype as well. In contrast, analogous glial progenitor cells were best identified in adult brain by the A2B5 phenotype alone, as adult GPCs largely down-regulated PDGF α R expression with maturation, despite retaining the functional characteristics of GPCs. Our approach recognized that the transcriptional hallmarks of hGPC identity may evolve with maturation and aging, such that the nominally canonical markers of the GPC phenotype at one point in differentiation may not be so at an earlier of later stage.

Some of the terminology used is not entirely clear. As stated in the introduction – “Glial

progenitor cells (GPCs, also referred to as oligodendrocyte progenitor cells or NG2 cells) emerge during the 2nd trimester to colonize the human brain, and persist in abundance throughout adulthood". The A2B5+ cells isolated from adults have little PDGFR ("PDGFRA in adult A2B5+ hGPCs was expressed with a median TPM of only 0.55, compared to its median TPM of 47.56 in fetal A2B5+ cells). One asks whether these adult A2B5+ cells referred to by some as pre-oligodendrocytes are not the same population as GRCs. As indicated GPCs are present in the adult human brain but have not been successfully isolated as viable cells for functional studies.

I'm sorry if this point remains a bit confusing. GPCs are present in the adult human brain, and have been isolated as such using A2B5; we first did so using a CNP2 promoter-driven reporter strategy (Roy et al., J. Neurosci., 1999; PMID: 10559406), which revealed that CNP2:GFP+ cells sorted from the adult human brain expressed A2B5, and accordingly, that mitotic oligodendrocyte progenitors could be isolated from the adult human brain using A2B5-based FACS. The emphasis here is on mitotic – the pre-OPCs first described by Armstrong and colleagues were post-mitotic; the adult-derived A2B5-defined population is mitotic, and manifests oligodendrocytic production analogous to that of fetal CD140-derived GPCs, both in vitro and in vivo (Windrem et al., Nature Medicine, 2004; PMID: 14702638). As such, these cells are functionally homologous to OPCs, which we prefer both herer and elsewhere to call GPCs, to accommodate their bilineage oligodendrocyte-astrocyte potential. The practical issue here is that A2B5 also recognizes young neurons in early development, hence the double sorting of fetal brain using PSA-NCAM, as an exclusionary marker for neuroblasts. To make this point clearer, we have added the following text and citations to the **first paragraph of the Results**:

Since A2B5 also recognizes young neurons in early development, we used two-color FACS to immunodeplete fetal brain samples for PSA-NCAM at the time of A2B5 enrichment, so as to exclude contaminating neuroblasts.; hence the definition of this fetal hGPC pool as A2B5+/PSA-NCAM-

The majority of ipsc cells studied, as stated in the rebuttal letter, were PGDFR+ indicating they were early in the lineage – the study did not intend to address a comparison of early and late progenitors to match the analysis of primary fetal PDGFR and adult A2B5 cells. The ipsc studies do not address properties of cells from old and young donors.

Yes, the referee is correct that iPSC-derived hGPCs, as newly generated cells, have a fetal-like phenotype - at least over the in vitro range (140-160 DIV) that we report here. In this paper, our iPSC studies were intended only to assess the ability of aging-associated TFs to prematurely age these otherwise young GPCs. Hence, these were ideal cells at an appropriate state for us to assess the ability of ZNF274, E2F6 and other aging-associated GPC TFs to drive the aged phenotype – which they did, decidedly so. We'd already pointed this out in the Introduction (page 2), as follows:

"We found that over-expression of these adult repressors in newly generated human iPSC-derived GPCs, which are analogous to fetal hGPCs in their expression signatures, induced transcriptional signatures that substantially recapitulated those of adult hGPCs..."

Perhaps most of the above concerns can be addressed in the discussion with a focus on the fundamental regulatory pathways identified and leave open the question how they impact on each cell type during human aging.

We thank the referees for their comments, and trust that the additional explanations that we have added to both the Results and Discussion have significantly improved the clarity of the manuscript, and thus the strength of its fundamental message, that it is the coordinate

activation of a set of repressors of developmental transcription that drives the aging of adult human glial progenitor cells.